# BOLT: Decision-Aligned Distillation and Budget-Aware Routing for Constrained Multimodal QA on Robots

**Tengjun Ni[1], Xin Yuan[2,3], Shenghong Li[2], Kai Wu[1], Ren Ping Liu[1], Wei Ni[4], Wenjie Zhang[3]**
[1]University of Technology Sydney, Sydney, Australia
[2]Data61, Commonwealth Scientific and Industrial Research Organization (CSIRO), Sydney, Australia
[3]University of New South Wales, Sydney, Australia
[4]Edith Cowan University, Perth, Australia
`{Tengjun.Ni, Kai.Wu, RenPing.Liu}@uts.edu.au`
`{xin.yuan, wei.ni}@ieee.org`
`shenghong.li@csiro.au, wenjie.zhang@unsw.edu.au`

## Abstract

Robotic systems can require multimodal reasoning under stringent constraints of latency, memory, and energy. Standard instruction tuning and token-level distillation fail to deliver decision quality, reliability, and interpretability under these constraints. We introduce BOLT, a decision-aligned distillation and budget-aware routing framework that treats multi-choice prediction as a decision surface to be aligned during training and selectively refined at inference. During training, BOLT introduces Option-level Decision Distillation to align student models directly on the decision surface of multi-choice answers, thereby eliminating prompt artifacts, improving calibration, and optimizing the exact output space. At inference, BOLT activates Budget-aware Test-time Augmentation, a calibrated router that uses low-cost signals such as confidence, margin, entropy, retrieval affinity, and agreement across short question decompositions to trigger high-resolution reevaluation, type-matched retrieval exemplars, or question decomposition only when their expected benefit outweighs cost. On Robo2VLM-1, a 2B BOLT student distilled from LLaVA-1.5-13B improves accuracy from 28.66 in zero-shot to 42.89 with decision distillation and to 50.50 with budgeted routing, surpassing the 13B teacher at 36.74. It lowers expected calibration error, strengthens the risk-coverage frontier, and slashes GPU memory from 26,878 MB for the teacher to 3,035 MB for the distilled student, and 3,817 MB with all augmentations enabled. By constraining outputs to valid options while exposing retrieved evidence and decomposition traces, BOLT reduces hallucination and provides transparent decision-making, enabling large-model quality on edge robots.

## 1 Introduction

Multimodal foundation models have progressed rapidly from contrastive vision-language pretraining to instruction-following vision-language models capable of grounded reasoning and multi-step perception (Radford et al., 2021; Jia et al., 2021). Systems, such as LLaVA (Liu et al., 2023) and Qwen2-VL (Wang et al., 2024), demonstrate strong zero-/few-shot performance across diverse visual question answering (VQA) tasks. There is growing interest in pushing these capabilities onto robots and embedded platforms. In parallel, many robotics benchmarks adopt constrained-output formulations (e.g., colors, arrow directions, options A–E, yes/no), which enable deterministic interfaces and safety checks and are well suited to on-device control and real-time loops (Brohan et al., 2022; Chen et al., 2025; Gordon et al., 2018; Teney et al., 2018).

A practical challenge arises: achieving the decision quality of large vision-language models (VLMs) for constrained, multi-choice decision making while respecting strict latency, memory, and energy budgets on edge hardware. Prior work takes several approaches to this goal. Token-level knowledge

distillation (KD) inherited from text-only language models (LMs) seeks to transfer teacher behavior at the sequence level (Hinton et al., 2015; Kim & Rush, 2016). Compact VLMs are directly fine-tuned on instruction-following data to better conform to task prompts. Always-on test-time enhancements such as higher-resolution re-evaluation and retrieval-augmented prompting with same-domain exemplars aim to boost accuracy (Lewis et al., 2020; Rubin et al., 2021).

Selective prediction with budgeted or dynamic inference, covering confidence-based abstention (Geifman & El-Yaniv, 2017) as well as early-exit and adaptive-compute methods (Teerapittayanon et al., 2016; Figurnov et al., 2017), trades coverage for risk under resource constraints. Parameter-efficient adapters like LoRA and quantization-aware QLoRA further reduce adaptation cost and memory (Hu et al., 2022; Dettmers et al., 2023). Beyond raw accuracy, many strategies pursue better interpretability and reduced hallucination, including retrieved-evidence provenance, decomposition traces, and calibration, but most such methods incur substantially higher compute, memory, or latency when applied uniformly. However, optimizing explicitly for multi-choice decision surfaces remains limited, and few studies simultaneously improve decision accuracy, increase interpretability, and mitigate hallucination under tight on-device resource constraints.

These gaps manifest in constrained, multi-choice robotic perception as several persistent pain points. Token-level distillation aligns surface form under a particular prompt template, not the decision surface over the option set used in constrained decoding, which can make the student brittle and misaligned with evaluation. Always-on enhancements improve accuracy but increase latency and energy consumption, violating tight budgets; naive decomposition procedures may introduce spurious intermediate steps that diverge from visual evidence (Kim et al., 2020). Compact VLMs are commonly under-calibrated (Guo et al., 2017), undermining selective computation and abstention.

Hallucination remains nontrivial in small models and is compounded by limited interpretability: it is often unclear why a decision was taken or which evidence supported it (Li et al., 2023b). Existing selective or budgeted inference seldom couples uncertainty with retrieval affinity or with agreement across decompositions, and evaluations rarely report risk–coverage or accuracy–budget frontiers for constrained multimodal QA. Panel-based layouts and tiny colored markers exacerbate small-model failures, while real-time control imposes per-frame budgets and VRAM ceilings. Taken together, these limitations leave optimization for the multi-choice decision surface underdeveloped; under tight on-device budgets, few methods simultaneously improve decision accuracy, increase interpretability, and mitigate hallucination.

**Contributions** We address the above-mentioned limitations with a decision-centric strategy that aligns the student with the teacher at the level of answer options and allocates additional test-time compute only when inexpensive signals indicate positive expected gain under a target budget. The training component performs decision-aligned distillation at the option level so that the student learns the teacher's preference over candidate answers under constrained decoding. The inference component uses budgeted, risk-calibrated routing to decide whether to re-evaluate at higher resolution, augment with retrieved same-domain exemplars, or invoke Question Decomposition (QD). This design not only improves accuracy-budget tradeoffs and calibration, but also mitigates hallucination by constraining outputs to valid options and grounding with retrieved context, and enhances interpretability by exposing decomposition traces and the retrieved exemplars that inform decisions.

We introduce Budgeted Option-Level Transfer (BOLT), a decision-centric framework for constrained, multiple-choice VQA on robots.[1] BOLT treats multiple-choice prediction as a decision surface to be aligned and then selectively refined: training performs option-level distillation to match teacher-student preferences over answers, while inference uses a budgeted, risk-calibrated router that spends extra compute only when inexpensive signals suggest positive expected gain (e.g., high-resolution re-evaluation, type-matched retrieval, short QD). By unifying decision alignment with selective computation, BOLT achieves large-model decision quality under tight on-device latency/memory/energy budgets, delivering better accuracy-budget and risk-coverage trade-offs, sharper calibration, fewer hallucinations through constrained outputs and grounding, and clearer interpretability via visible evidence traces.

The key contributions of this paper are summarized as follows.

---

[1]https://github.com/A-leyenda/BOLT

- Budgeted Option-Level Transfer (BOLT): A decision-centric framework for constrained, multi-choice VQA on robots that unifies option-level decision distillation (ODD) with budgeted test-time augmentation (bTTA). By aligning training and inference around the multi-choice decision surface and spending compute only when inexpensive signals predict benefit, BOLT attains large-VLM decision quality under tight on-device latency/memory/energy budgets, improving accuracy-budget and risk-coverage frontiers and calibration while reducing hallucination and increasing interpretability.

- Option-level Decision Distillation: A decision-aligned objective that matches teacher-student option distributions derived from answer-segment likelihoods, improving Exact-Match and calibration over token-level distillation on constrained multimodal QA.

- Budgeted Test-time Augmentation: A risk-calibrated router that adapts inference compute (Hi-Res, retrieval augmentation, QD) per instance using uncertainty and retrieval-affinity features; under mild monotonicity assumptions, this induces a near-threshold policy that optimizes accuracy subject to a compute budget and empirically improves risk-coverage and accuracy-budget frontiers.

- Mitigating hallucination and improving interpretability: By constraining outputs to valid option sets, grounding predictions with retrieved exemplars, and exposing QD traces, the framework reduces contradiction-to-image errors and invalid-option responses while providing human-inspectable evidence chains; quantitative analysis appears in Section 4.6.

Although our empirical study is conducted on a single panel-style robotic VQA benchmark, the setting it represents is not merely a toy benchmark. In many deployed robotic systems, perception and planning stacks reduce the world to a small set of discrete options, such as candidate grasps, poses, waypoints, high-level skills, or yes/no checks, and controllers are required to choose exactly one option under tight safety and latency constraints. Large-scale public datasets with this constrained, multimodal decision interface are still relatively scarce, especially in robotics, but the underlying pattern of option-based decisions is pervasive in industrial and household robots. Our focus on constrained multi-choice QA should therefore be understood as modeling a realistic decision interface rather than overfitting to a single benchmark.

## 2 RELATED WORK

### 2.1 VISION-LANGUAGE MODELS FOR ROBOTICS AND CONSTRAINED-OUTPUT QA

Early vision-language pretraining focused on contrastive objectives that align images and texts in a shared embedding space (Radford et al., 2021; Jia et al., 2021). Subsequent instruction-following VLMs integrate autoregressive decoding and multimodal instruction tuning (e.g., LLaVA (Liu et al., 2023), Qwen2-VL (Wang et al., 2024), BLIP-2 (Li et al., 2023a)), enabling strong zero-/few-shot generalization on open-ended and task-driven QA. In robotics, there is an increasing emphasis on models that interface with perception and control stacks under strict latency and energy constraints (e.g., RT-1/RT-2) (Brohan et al., 2022; Zitkovich et al., 2023). Many robotic and diagnostic VQA settings adopt constrained-output formulations to facilitate deterministic interfaces, safety checks, and reliable evaluation (Chen et al., 2025; Gordon et al., 2018; Teney et al., 2018). However, this regime exposes a specific gap: most training and adaptation practices remain token-level and open-ended, which misaligns with the option-based decision surface used at evaluation and tends to increase compute when applied uniformly on devices. We therefore study an option-aligned alternative and show that aligning option distributions better preserves decision quality for constrained decoding under on-device budgets.

### 2.2 KNOWLEDGE DISTILLATION FOR MULTIMODAL MODELS AND CONSTRAINED DECISIONS

Knowledge distillation (KD) was introduced as logit-matching with temperature scaling (Hinton et al., 2015) and later extended to sequence-level distillation for text generation (Kim & Rush, 2016). In vision and NLP, distillation spans token/logit-based matching, response-level training, and feature/attention transfer, often improving latency and memory without fully retaining calibration or decision boundaries (Sanh et al., 2019; Touvron et al., 2021). Multimodal KD for VQA

typically mirrors token-level or cross-entropy supervision, which can entangle prompt-template idiosyncrasies with answer learning and may misalign with the constrained option space used at evaluation. Parameter-efficient tuning reduces adaptation cost and memory footprint (Hu et al., 2022; Dettmers et al., 2023), but does not by itself address decision alignment or budgeted test-time compute. Progress in distillation and parameter-efficient tuning notwithstanding, faithfully transferring a teacher's option-level decision quality and calibration to compact VLMs for constrained multiple-choice decoding without overrunning on-device memory budgets remains elusive.

## 2.3 RETRIEVAL-AUGMENTED INFERENCE AND BUDGETED/DYNAMIC COMPUTE

Retrieval-augmented methods provide external evidence or exemplars to improve factuality and domain transfer, including RAG-style retrieval+generation, Fusion-in-Decoder, and memory-augmented language models (Lewis et al., 2020; Izacard & Grave, 2020; Borgeaud et al., 2022; Khandelwal et al., 2019; Rubin et al., 2021). In vision-language QA, retrieval can supply task-type exemplars for in-context guidance, but naive always-on use increases latency and energy and can degrade reliability when irrelevant evidence is injected. Separately, budgeted or dynamic inference studies how to adapt computation to instance difficulty: early-exit and conditional computation in CNNs/Transformers, adaptive-depth/width routing, and selective prediction with abstention trade off coverage for risk under resource constraints (Teerapittayanon et al., 2016; Figurnov et al., 2017; Geifman & El-Yaniv, 2017). What remains absent is a cost-aware controller that, under an explicit compute budget, relies on trustworthy instance-level signals to trigger high-cost augmentations only when they deliver positive net benefit.

## 2.4 CALIBRATION, HALLUCINATION, AND STRUCTURED DECOMPOSITION

Modern neural networks are often miscalibrated; post-hoc temperature scaling partially remedies this, but can be unstable across domains (Guo et al., 2017; Minderer et al., 2021). In VLMs, multi-modal hallucination persists (e.g., contradictions to the image or invalid-option outputs), and measuring/mitigating it remains an active area (Li et al., 2023b; Rohrbach et al., 2018). Interpretability tools range from retrieved-evidence provenance to structured reasoning traces. Modular/structured approaches, including neural module networks, program-like reasoning, and decomposition-style prompting, seek to expose intermediate structure and reduce spurious correlations (Yi et al., 2018; Hudson & Manning, 2018; Zhou et al., 2022). Despite techniques for post-hoc calibration, hallucination mitigation, and structured decomposition, it remains difficult to simultaneously improve accuracy, calibration, and hallucination while preserving tight on-device budgets and avoiding uniform overhead.

# 3 METHODOLOGY

## 3.1 PROBLEM SETUP AND NOTATION

We study constrained-output VQA for robotics. Each example is denoted as $(x, q, \mathcal{O}, y)$, where $x$ is an image or panel layout, $q$ is a natural-language question, $\mathcal{O} = \{o_1, \ldots, o_K\}$ is a finite option set, and $y \in \{1, \ldots, K\}$ is the ground-truth index set. We use a large teacher VLM $T$ and a compact student VLM $S_\theta$ with parameters $\theta$. Both are evaluated under constrained decoding: the model must output exactly one option text from $\mathcal{O}$.

We fix a chat template that places $(x, q)$ in a user turn and the answer in an assistant turn. For option $o_k$, let the tokenized answer be $\mathbf{a}^{(k)} = (a_1^{(k)}, \ldots, a_{L_k}^{(k)})$, and let the full sequence be $\mathbf{z}^{(k)} = (z_{1:L_0}^{(k)}, \mathbf{a}^{(k)})$ where indices $\mathcal{A}^{(k)} = \{L_0 + 1, \ldots, L_0 + L_k\}$ correspond to the answer segment. For a model $M$, denote its next-token distribution by $p_M(\cdot \mid \mathbf{z}_{<t})$.

**Answer-segment likelihood.** We deliberately score only the assistant answer segment:

$$s_M(k \mid x, q) := \sum_{t \in \mathcal{A}^{(k)}} \log p_M\left(a_{t-L_0}^{(k)} \mid \mathbf{z}_{<t}^{(k)}\right). \tag{1}$$

This removes prompt-template wording from supervision and focuses exactly on the part that is evaluated under constrained decoding.

Figure 1: **Pipeline overview.** *Left: Distill phase (ODD).* The teacher VLM provides option-level supervision via answer-segment likelihoods; the student learns to match the teacher's decision distribution and is then calibrated. *Right: Inference phase (bTTA).* A fast constrained pass produces confidence features; a router selectively triggers **HR** re-evaluation, **tmRAG**, and **QD** and combines their outputs to form the final prediction under a compute budget.

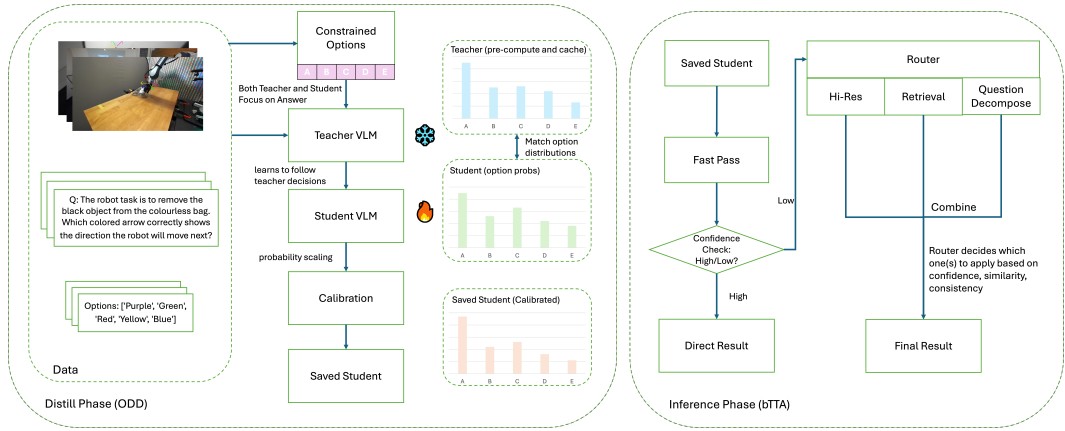

**Roadmap to test time.** Having defined how decisions are formed under constrained decoding, we first train the student so that its *option-level* decisions match the teacher, then design a budgeted router that decides when to pay for costlier augmentations at inference.

## 3.2 System Overview and Design

As shown in Fig.1, we design BOLT, a decision-centric framework for constrained multimodal QA on robots that delivers large-model decision quality at small-model cost.

During training, BOLT utilizes Option-level Decision Distillation (ODD) to match the teacher's temperature-softmax over answer options, which are computed from answer-segment likelihoods. A LoRA/QLoRA student is optimized with a small cross-entropy anchor, aligning the constrained-decoding decision surface, improving calibration, and mitigating prompt and length artifacts.

During inference, a budgeted router reads lightweight signals from the student distribution augmented by type-matched retrieval affinity and agreement across short QDs. It triggers only the helpful augmentations under a budget, namely HR high-resolution re-evaluation, tmRAG type-matched retrieval exemplars, and QD. This design concentrates computation where it pays off, yielding stronger accuracy-budget and risk-coverage tradeoffs than always-on enhancements. It reduces invalid-option and image-contradiction errors by grounding predictions, and improves interpretability via retrieved evidence and decomposition traces. Temperature scaling further calibrates probabilities used for routing.

### 3.2.1 Option-level Decision Distillation (ODD)

To transfer the decision quality of a large teacher to a compact student under constrained decoding, we propose ODD, a decision-aligned objective that supervises the model at the level of answer options rather than tokens. Token-level KD from text LMs mixes prompt and answer tokens, penalizes template wording differences that are irrelevant at evaluation, and encourages surface-form imitation; in constrained QA, the evaluation hinges on the teacher's preference over the option set. ODD therefore scores, for each option, only the assistant's answer segment, sums its token log-likelihoods to form per-option scores, converts them with a temperature-softmax into a teacher option distribution, and trains the student to match this distribution with a KL term plus a small cross-entropy to the ground-truth option. This directly targets the decision surface realized by constrained decoding, improves calibration, and avoids the prompt-answer tug-of-war inherent to token-level distillation; when option strings differ markedly in length, a light length-bias correction can be applied.

**Option distributions and decision-aligned loss.** We first turn the teacher's answer-segment scores into a calibrated preference over options by applying a temperature-softmax, which smooths overconfident peaks and exposes relative utilities across $\mathcal{O}$:

$$p_T(k \mid x, q) = \frac{\exp\big(s_T(k)/\tau_{\mathrm{kd}}\big)}{\sum_{j=1}^{K} \exp\big(s_T(j)/\tau_{\mathrm{kd}}\big)}, \qquad \tau_{\mathrm{kd}} > 0. \tag{2}$$

We cache $\{p_T(k)\}$ offline for all training items so that student training compares against a fixed teacher distribution without repeatedly querying the teacher. This cache is used only during offline training; at inference time the deployed system never queries the teacher and relies solely on the distilled student. To make the teacher and student directly comparable in the same probability simplex, we build the student's option distribution by normalizing its own answer-segment scores using the same softmax functional form (but without temperature scaling):

$$p_S(k \mid x, q; \theta) = \frac{\exp\big(s_\theta(k)\big)}{\sum_{j=1}^{K} \exp\big(s_\theta(j)\big)}, \qquad s_\theta(k) = s_{S_\theta}(k \mid x, q). \tag{3}$$

This places both models' decisions on a common, prompt-invariant option space that mirrors constrained decoding at evaluation.

With $p_T$ and $p_S$ defined, we optimize a decision-aligned objective that pulls the entire student distribution toward the teacher while retaining a minimal anchor to ground truth:

$$\mathcal{L}_{\mathrm{ODD}}(\theta) = \lambda_{\mathrm{KL}} \, \mathrm{KL}(p_T \,\|\, p_S) \; + \; \lambda_{\mathrm{CE}} \, \mathrm{CE}(\delta_y \,\|\, p_S), \qquad \lambda_{\mathrm{KL}}, \lambda_{\mathrm{CE}} \geq 0. \tag{4}$$

Intuitively, the KL term shapes the student's decision surface by matching the teacher's option preferences, and the small CE term provides a ground-truth anchor that stabilizes learning, preserves rare-option recall, and corrects teacher bias in ambiguous cases.

**Invariances, length bias, and gradient shape.** ODD operates on sums of answer–token log-probabilities, is invariant to adding any constant to all $s_\theta(k)$, and is robust to benign tokenization changes for fixed option strings. This robustness comes from the fact that splitting or merging tokens within an option only changes how the sequence log-likelihood is decomposed over time; the total sum $\sum_{t \in \mathcal{A}^{(k)}} \log p(a_t^{(k)} \mid \mathbf{z}_{<t}^{(k)})$ remains a consistent measure of that option's probability under a given model, and our loss always compares options within the same model and tokenization. When option strings differ substantially in length, we correct the scores by

$$\widetilde{s}_\theta(k) = s_\theta(k) - \gamma \log L_k \quad \text{or} \quad s_\theta(k)/L_k, \qquad \gamma \in [0, 1]. \tag{5}$$

For intuition about decision alignment, let $\mathbf{s}_\theta = [s_\theta(1), \dots, s_\theta(K)]^\top$. The gradients of Eq. (4) with respect to $\mathbf{s}_\theta$ are given by

$$\nabla_{\mathbf{s}_\theta} \mathrm{KL}\big(p_T \| p_S\big) = \mathbf{p}_S - \mathbf{p}_T, \qquad \nabla_{\mathbf{s}_\theta} \mathrm{CE}\big(\delta_y \| p_S\big) = \mathbf{p}_S - \delta_y,$$

so the total signal pushes the student option distribution toward the teacher and the ground-truth anchor. When backpropagated to token logits, this supervision is applied only to answer-segment tokens, avoiding the prompt–answer tug-of-war that plagues token-level distillation.

**Parameter-efficient training.** We train LoRA adapters in attention and MLP projections (optionally the multimodal projector) while keeping the base quantized with QLoRA. ODD gradients flow only through adapter paths and the projector, enabling single-GPU training.

### 3.2.2 BUDGETED TEST-TIME AUGMENTATION (BTTA)

To maximize accuracy under an explicit compute budget while preserving low latency on edge hardware, we propose bTTA, an adaptive inference-time framework that allocates computation per instance. The distilled student first performs a fast constrained pass, from which we derive a compact routing feature vector based on the option distribution and auxiliary cues. A learned policy decides whether to execute high-resolution (HR) re-evaluation, type-matched retrieval exemplars (tmRAG), or Question Decomposition (QD). Each augmentation is modeled as an action with measurable cost and a learned success probability; bTTA triggers actions only when the predicted gain exceeds the cost within the budget, yielding calibrated final decisions and improved accuracy-budget tradeoffs.

**Routing features and policy.** From the pass-1 option distribution $p_S$ we compute confidence, margin, and entropy,

$$p_{\max} = \max_k p_S(k), \qquad \Delta = p_{(1)} - p_{(2)}, \qquad H = -\sum_k p_S(k) \log p_S(k).$$

We augment them with a *retrieval affinity*

$$\rho = \frac{1}{K_r} \sum_{j \in \text{Top-}K_r} \cos\big(\phi(x,q), \phi(x_j, q_j)\big),$$

where the memory stores only same-type items to avoid cross-type interference and, with an agreement score across short QD, runs

$$\kappa = 1 - \frac{2}{K_d(K_d - 1)} \sum_{k < k'} \text{JS}\big(p_S^{(k)} \,\|\, p_S^{(k')}\big),$$

which increases when independent decompositions concur. The feature vector $\mathbf{f} = [p_{\max}, \Delta, H, \rho, \kappa]$ drives the router. Let actions $\mathcal{A} = \{\text{HR}, \text{RAG}, \text{QD}\}$ collect HR, tmRAG, and QD. Each action $a$ has a normalized marginal cost $C_a$ and a binary improvement label $\Delta\text{Acc}_a$ relative to the current prediction. A gain model $g_\omega(\mathbf{f}, a) \approx \Pr[\Delta\text{Acc}_a = 1 \mid \mathbf{f}]$ is learned on validation logs, and the per-instance decision solves

$$\max_{\alpha_a \in \{0,1\}} \sum_{a \in \mathcal{A}} \alpha_a \, g_\omega(\mathbf{f}, a) \, W_a \quad \text{s.t.} \quad \sum_{a \in \mathcal{A}} \alpha_a \, C_a \leq B, \tag{6}$$

with small empirical weights $W_a$ and budget $B$. Its Lagrangian relaxation yields a simple near-threshold rule

$$\text{trigger } a \iff g_\omega(\mathbf{f}, a) \, W_a \geq \tau \, C_a, \quad \text{with cumulative cost} \leq B, \tag{7}$$

which activates an augmentation when predicted gain exceeds cost by a threshold $\tau$ tuned on validation to satisfy the average budget and maximize accuracy. With diminishing returns across actions, greedy selection by this net value is a strong approximation.

**Actions and calibration.** *HR* replaces the image with a larger short-edge for a second constrained pass to recover fine detail. *tmRAG* retrieves Top-$K_r$ same-type exemplars $(\text{desc}, q, a)$ by cosine similarity in an encoder space and appends them to the prompt; the student re-answers to obtain $p_S^{\text{RAG}}$. *QD* elicits $K_d$ short decompositions (two-three checks) with diversity via seeds or few-shot permutations, producing distributions $\{p_S^{(k)}\}_{k=1}^{K_d}$ and an aggregated vote

$$\hat{p}_S(\cdot) = \frac{1}{K_d} \sum_{k=1}^{K_d} p_S^{(k)}(\cdot), \qquad \hat{o} = \arg\max_o \hat{p}_S(o), \tag{8}$$

while the agreement $\kappa$ feeds back into routing to suppress unhelpful decompositions. Because routing relies on probabilities, we apply temperature scaling on a validation split,

$$p_S^{\text{cal}}(k) = \frac{\exp\big(s_\theta(k)/\tau_{\text{cal}}\big)}{\sum_j \exp\big(s_\theta(j)/\tau_{\text{cal}}\big)},$$

and reuse calibrated $p_{\max}$, margin and entropy in Appendix F.

# 4 EXPERIMENTS

## 4.1 EXPERIMENT SETUP

We conduct the main study on **Robo2VLM-1** (Chen et al., 2025), a panel-style robotic perception QA benchmark with constrained option sets. Following our constrained decoding interface (Sec. 3), we evaluate by *Accuracy (Acc)* over options. For Robo2VLM-1, we form three non-overlapping splits by unique image identifiers: `train-kd` (used only to build the teacher option-distribution cache), `val` (router calibration and temperature scaling), and `test` (final reporting). Retrieval

memories and decomposition exemplars are constructed exclusively from `train-kd` to avoid leakage into `val`/`test`. Unless otherwise stated, all results are single-image, batch size 1.

We distill from three teachers spanning families and sizes, Qwen2.5-VL-7B, LLaVA-1.5-7B, and LLaVA-1.5-13B, using their per-option answer-segment likelihoods to form temperature-softmax teacher distributions $p_T$ for ODD. The student is Qwen2-VL-2B-Instruct, trained with ODD via LoRA/QLoRA. To assess cross-architecture generality, we also report results for a PaliGemma-2-3B student distilled from LLaVA-1.5-13B. All models share the same constrained-decoding interface and bTTA configuration for fair comparison. Unless otherwise specified, we report each teacher-student setting with $\tau_{\mathrm{kd}}$ and the CE weight tuned on `val`.

To isolate the effect of decision-aligned distillation, we also implement a strong token-level KD baseline. Specifically, the teacher's next-token logits over the full answer sequence (including both prompt and answer tokens) supervise the student via a temperature-scaled KL term plus a standard cross-entropy on ground-truth tokens. In all result tables, rows marked as "Token-KD" correspond to this baseline; rows marked as "ODD" replace token-level supervision with our option-level decision distillation while keeping all other settings identical.

## 4.2 Main Results

Table 2 summarizes accuracy under a unified constrained-decoding protocol. A 2B student trained with option-level decision distillation attains 42.89% when distilled from LLaVA-1.5-13B, exceeding the 13B teacher at 36.74% and the 2B zero-shot baseline at 28.66% by sizable margins. Adding the budgeted test-time augmentation policy further increases accuracy to 50.50, while maintaining the same decoding interface. Replacing ODD with token-level KD yields lower accuracy across teachers: 33.91%/36.21%/37.58% when distilled from LLaVA-7B/Qwen2.5-VL-7B/LLaVA-13B, trailing ODD by 4.62/4.31/5.31 points, respectively. With bTTA, token-level KD improves to 39.92/44.42/47.02 but still lags behind ODD+bTTA at 45.43%/47.16%/50.50%. The pattern is consistent across other teachers: distillation from LLaVA-1.5-7B yields 38.53% and rises to 45.43% with the policy, and distillation from Qwen2.5-VL-7B yields 40.52% and rises to 47.16%. These results indicate that aligning decisions at the option level closes most of the capacity gap, and instance-adaptive compute converts residual uncertainty into additional accuracy without changing the model architecture or the evaluation protocol. For completeness, NLL, Brier, ECE, and AURC, together with the full ablation breakdown, are reported in Appendix Table 9.

## 4.3 Ablations

Table 1 examines the contribution of each inference-time component to the student distilled from LLaVA-1.5-13B. High-resolution re-evaluation primarily fixes resolution-limited errors and moves accuracy from 42.89% to 46.64%. Type-matched retrieval adds domain-appropriate exemplars and moves accuracy to 44.31% when used alone, and to 48.25% when combined with high resolution. QD reduces reasoning variance on difficult cases and moves accuracy to 45.47% alone, and to 48.92% together with high resolution. Enabling all three components achieves 50.50%. The gains are monotone and nearly additive, supporting the design of a router that triggers only the actions whose predicted gain exceeds their cost.

## 4.4 Budgeted Routing: Accuracy-Budget Behavior and Baselines

We study how the budgeted router allocates test-time compute and how this impacts accuracy under an explicit average budget $B$ for *ODD+bTTA*. Each optional augmentation is treated as a binary action with fixed per-trigger costs $(C_{\mathrm{HR}}, C_{\mathrm{tmRAG}}, C_{\mathrm{QD}}) = (0.50, 0.30, 0.35)$, on top of a base constrained pass costing $1.00$. Sweeping a calibrated threshold yields a discrete accuracy-budget frontier (Appx. Fig. 10). Accuracy improves monotonically from the single-pass ODD baseline at $B=1.00$ to our full-budget setting around $B\approx2.00$, where gains saturate.

Trigger compositions increase smoothly with $B$ (Appx. Table 4). Feature-importance diagnostics on the learned gain model show that HR is primarily gated by entropy $H$, tmRAG by same-type retrieval affinity $\rho$, and QD by agreement $\kappa$ across short decompositions, consistent with the router design in Sec. 3.2.2.

Table 1: Module ablations on Robo2VLM-1 (Acc %). Columns indicate whether **HR** (High-Resolution), **tmRAG** (type-matched retrieval exemplars), and **QD** (Question Decomposition) are enabled.

| Variant | HR | tmRAG | QD | Acc (%) |
|---|---|---|---|---|
| Qwen2 VL-2B (zero-shot) | N | N | N | 28.66 |
| Qwen2 VL-2B *distilled by* LLaVA-13B (ODD) | N | N | N | 42.89 |
| + tmRAG | N | Y | N | 44.31 |
| + QD | N | N | Y | 45.47 |
| + HR | Y | N | N | 46.64 |
| + HR + tmRAG | Y | Y | N | 48.25 |
| + HR + QD | Y | N | Y | 48.92 |
| + HR + tmRAG + QD | Y | Y | Y | **50.50** |

Table 2: Comparison on Robo2VLM-1 (Acc %). All methods share the same constrained decoding and prompt template.

| Model | Params (B) | Acc (%) |
|---|---|---|
| LLaVA 1.5-7B (zero-shot) | 7 | 21.58 |
| LLaVA 1.5-13B (teacher, zero-shot) | 13 | 36.74 |
| Qwen2 VL-2B (zero-shot) | 2 | 28.66 |
| *Teacher: LLaVA-1.5-7B → Student: Qwen2 VL-2B* | | |
| Qwen2 VL-2B *distilled by* LLaVA-7B (Token-KD) | 2 | 33.91 |
| + bTTA | 2 | 39.92 |
| Qwen2 VL-2B *distilled by* LLaVA-7B (Ours, ODD) | 2 | 38.53 |
| + bTTA | 2 | 45.43 |
| *Teacher: Qwen2.5-VL-7B → Student: Qwen2 VL-2B* | | |
| Qwen2 VL-2B *distilled by* Qwen2.5-VL-7B (Token-KD) | 2 | 36.21 |
| + bTTA | 2 | 44.42 |
| Qwen2 VL-2B *distilled by* Qwen2.5-VL-7B (Ours, ODD) | 2 | 40.52 |
| + bTTA | 2 | 47.16 |
| *Teacher: LLaVA-1.5-13B → Student: PaliGemma2-3B* | | |
| PaliGemma2-3B *distilled by* LLaVA-13B (Token-KD) | 2 | 35.50 |
| + bTTA | 2 | 40.21 |
| PaliGemma2-3B *distilled by* LLaVA-13B (Ours, ODD) | 2 | 39.64 |
| + bTTA | 2 | 45.92 |
| *Teacher: LLaVA-1.5-13B → Student: Qwen2 VL-2B* | | |
| Qwen2 VL-2B *distilled by* LLaVA-13B (Token-KD) | 2 | 37.58 |
| + bTTA | 2 | 47.02 |
| Qwen2 VL-2B *distilled by* LLaVA-13B (Ours, ODD) | 2 | 42.89 |
| + bTTA | 2 | **50.50** |

Under matched budgets, we compare to two single-signal dynamic-compute baselines (HR-Threshold and one-branch Early-Exit). Our router yields $+\sim$1.4–2.2 points accuracy and lower ECE/AURC by deciding both whether to escalate and which augmentation to trigger per instance (Appx. Table 5; more details are provided in Appx. F, G).

## 4.5 System Footprint and Deployability

Appx. Table 11a reports GPU memory with batch size one. The distilled 2B student occupies 3,035 MB, which reduces memory by 88.7% relative to the 13B teacher at 26,878 MB and by 41.0% relative to the 2B zero-shot baseline at 5,144 MB. Activating the full set of augmentations raises the footprint to 3,817 MB, adding only 782 MB and remaining about seven times lighter than the 13B teacher. In conjunction with the accuracy improvements in Table 2, this footprint makes the pipeline practical for edge deployment, since the model fits within commodity GPU memory while delivering higher accuracy than a much larger teacher.

For the LLaVA-1.5-13B to Qwen2-VL-2B setting, the *end-to-end* mean latency per question is 8.97 s. Of this, the *bTTA* pipeline (routing + selectively triggered HR/tmRAG/QD) contributes only 1.52 s on average, leaving 7.45 s for the first constrained pass (ODD student). Thus, bTTA accounts for 16.9% of total latency, while the pass-1 accounts for 83.1%. The effective throughput is 6.69 items/min (0.11 QPS); see Table 11b.

For reference, the 13B LLaVA teacher itself achieves 36.74% accuracy on Robo2VLM-1 with an average per-question latency of 22.30 s and an energy cost of 8,697 J on the same hardware. Our full BOLT pipeline (ODD student + bTTA) reaches 50.50% accuracy with 8.97 s latency and 1,522 J per query, corresponding to roughly a 2.5× speedup and an 82.5% reduction in energy while improving accuracy by 13.8 points. This operating point is suitable for high-level robotic perception and planning loops, whereas strict real-time low-level control would require further engineering and additional optimization, which we leave for future work.

## 4.6 Hallucination Analysis

We assess hallucination under constrained decoding using six proxies (IOR, NOA, Flip, HO_mean_wrong, OCW@0.7, and augmentation-conditioned contradiction rates RCR/QDC). The constrained interface removes string-form hallucinations outright (IOR= 0), while decision-level misuse of the "None of the above" sentinel is substantially reduced from 1.08% (zero-shot) to 0.37% (ODD pass-1) and 0.22% (ODD+bTTA). The router actively corrects uncertain cases, 26.71% of labels flip relative to pass-1, yielding fewer over-confident mistakes (OCW@0.7

4.18%→0.27%→0.19%) despite a slight rise in average confidence on remaining errors (HO_mean_wrong 0.2678→0.2946). Augmentation-conditioned diagnostics localize residual risks: retrieved exemplars can conflict with image-consistent answers (RCR 21.73%), while gated QD is largely but not perfectly self-consistent (QDC 1.74%). BOLT eliminates string-level hallucinations by design, curbs NOA misuse, and suppresses the high-confidence error tail; remaining errors are primarily driven by retrieval quality and decomposition policy. More details are in Appendix H.

## 5  LIMITATIONS AND FUTURE WORK

Our evaluation focuses on a single panel-style robotic VQA benchmark (Robo2VLM-1), reflecting the broader limitation that existing robotic multi-choice VQA datasets are scarce. Although our experiments focus on a robotic perception benchmark, the ODD objective and the budgeted router are modality-agnostic and can be applied to other multimodal multi-choice QA settings where the answer space is relatively uniform and constrained, as long as these settings also care about latency or model size. The tmRAG component induces retrieval-driven hallucinations (RCR 21.73%), which is a common limitation of RAG pipelines when exemplars imperfectly match the query. In the future, we will (i) expand to multiple public datasets and report per-type risk-coverage and calibration, (ii) replace or complement tmRAG with conflict-aware exemplar filtering, visual-entailment style option verification, and retrieval-free self-consistency voting, (iii) optimize the router and student under compute-aware objectives (latency/energy) to improve calibration and robustness under distribution shift, and (iv) move beyond purely offline supervised distillation by exploring on-policy decision distillation for constrained VLMs using logged robot interactions and simulated rollouts.

## 6  CONCLUSION

This work reframes constrained multimodal question answering for robotics as the joint problem of option-level decision alignment and on-demand inference, and demonstrates that aligning the student to the teacher in the option space used at evaluation, then spending computation only when inexpensive signals predict benefit, yields a compact system that is both accurate and reliable under tight resource budgets. BOLT transfers the teacher decision surface through Option-level Decision Distillation and concentrates computation via a calibrated, budgeted router, leading to substantial gains in accuracy, calibration, and interpretability without changing the interface or inflating memory, with a 2B student exceeding a 13B teacher on Robo2VLM-1 and showing stronger risk-coverage and accuracy-budget trade-offs along with lower invalid-option and image-contradiction errors. Future extensions include generalizing option-level alignment to broader structured outputs and multi-step control, strengthening retrieval filtering and consistency modeling while optimizing the router and the student jointly, and making energy and latency explicit objectives in end-to-end training followed by comprehensive evaluation across diverse robotic tasks and public benchmarks to promote reproducible assessment and responsible deployment.

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

## A  ADDITIONAL DERIVATIONS AND DETAILS

### A.1  MASKED LIKELIHOOD ON ANSWER SEGMENT

Let $\mathbf{z}^{(k)} = (z_1, \ldots, z_L)$ and mask $m_t = \mathbb{I}[t \in \mathcal{A}^{(k)}]$. The masked NLL is

$$\mathcal{L}_{\mathrm{NLL}}^{(k)}(\theta) = -\sum_{t=1}^{L} m_t \log p_\theta(z_t \mid \mathbf{z}_{<t}) = -\sum_{t \in \mathcal{A}^{(k)}} \log p_\theta(z_t \mid \mathbf{z}_{<t}). \tag{9}$$

By definition $s_\theta(k) = -\mathcal{L}_{\mathrm{NLL}}^{(k)}(\theta)$. In practice, we set labels outside the answer segment to $-100$ to exclude them from loss.

### A.2  GRADIENTS FOR ODD AND TOKEN-LEVEL BACKPROP

Let $\mathbf{s}_\theta = [s_\theta(1), \ldots, s_\theta(K)]^\top$, $p_S = \mathrm{softmax}(\mathbf{s}_\theta)$. For fixed $p_T$, we have

$$\frac{\partial \, \mathrm{KL}(p_T \| p_S)}{\partial s_\theta(j)} = p_S(j) - p_T(j), \qquad \frac{\partial \, \mathrm{CE}(\delta_y \| p_S)}{\partial s_\theta(j)} = p_S(j) - \mathbb{I}[j=y]. \tag{10}$$

Thus

$$\frac{\partial \mathcal{L}_{\mathrm{ODD}}}{\partial s_\theta(j)} = \lambda_{\mathrm{KL}} \big( p_S(j) - p_T(j) \big) + \lambda_{\mathrm{CE}} \big( p_S(j) - \mathbb{I}[j=y] \big). \tag{11}$$

Each $s_\theta(j)$ is a sum of answer-token log-probs. Let $\boldsymbol{\ell}_t$ be pre-softmax logits and $y_t$ be the gold token at time $t$. For $t \in \mathcal{A}^{(j)}$,

$$\frac{\partial s_\theta(j)}{\partial \boldsymbol{\ell}_t} = \mathbf{e}_{y_t} - \mathrm{softmax}(\boldsymbol{\ell}_t), \quad \frac{\partial s_\theta(j)}{\partial \boldsymbol{\ell}_t} = \mathbf{0} \text{ for } t \notin \mathcal{A}^{(j)}. \tag{12}$$

Hence, ODD produces an *option-weighted* learning signal focused on answer tokens, unlike token-level KD, which distributes weight over prompt tokens as well.

### A.3 TEMPERATURE, LABEL SMOOTHING, AND LENGTH CORRECTION

Teacher temperature $\tau_{\mathrm{kd}}$ controls the softness of $p_T$; we select $\tau_{\mathrm{kd}} \in [1.5, 3]$. We optionally use label smoothing $\varepsilon$ in the CE term: $\mathrm{CE}((1-\varepsilon)\delta_y + \varepsilon/K \parallel p_S)$, stabilizing optimization when $p_T$ is sharp. Length correction in equation 5 mitigates rare cases where $L_k$ differs across options.

### A.4 ROUTER LEARNING AS CONTEXTUAL BANDITS

We cast routing as a contextual bandit with context $\mathbf{f}$ and actions $a \in \mathcal{A}$ producing binary reward $\Delta \mathrm{Acc}_a$ and cost $C_a$. We collect an offline log on a held-out split by executing pass-1, then all augmentations (for supervision only), recording $(\mathbf{f}, a, \Delta \mathrm{Acc}_a, C_a)$. We train a probabilistic classifier $g_\omega(\mathbf{f}, a) \approx \Pr[\Delta \mathrm{Acc}_a = 1 | \mathbf{f}]$ with class-imbalance weights and temperature calibration. At test time, we apply the near-threshold rule in equation 7. When logs are collected under a different policy, an unbiased estimator of the expected gain can be formed by inverse propensity scores (IPS): $\widehat{\mathbb{E}}[\Delta \mathrm{Acc}_a | \mathbf{f}] = \frac{\mathbb{I}[a=a_0] \Delta \mathrm{Acc}_a}{\pi(a_0 | \mathbf{f})}$, where $\pi$ is the logging propensity; we find that direct supervised $g_\omega$ is sufficiently stable in our setting.

### A.5 LAGRANGIAN VIEW, THRESHOLD POLICY, AND SEQUENTIAL AUGMENTATIONS

The constrained problem, i.e., (6), has Lagrangian $\mathcal{L}(\alpha, \tau) = \sum_a \alpha_a \left( \mathbb{E}[\Delta \mathrm{Acc}_a | \mathbf{f}] - \tau C_a \right) + \tau B$. For fixed $\tau$, the maximizer sets $\alpha_a = 1$ iff the net value is nonnegative, giving the near-threshold rule in (7). When applying multiple augmentations sequentially, if the expected improvement exhibits *diminishing returns* (submodularity) over the action set, the greedy selection that iteratively picks the largest positive net value achieves a $(1 - 1/e)$-approximation to the optimal set (Nemhauser bound). Empirically, we can cap the number of augmentation rounds (e.g., one or two) to keep latency predictable.

### A.6 AGREEMENT METRICS AND ALTERNATIVES

Beyond $\kappa$ via average pairwise Jensen–Shannon similarity, we consider: (i) majority-vote rate among $\{\arg\max p_S^{(k)}\}$; (ii) Kendall's $W$ on option rankings; (iii) variance of $\{p_S^{(k)}\}$ along the top eigenvector. We use $\kappa$ for differentiability in ablations and majority-vote for reporting.

### A.7 SELECTIVE RISK, RISK–COVERAGE, AND CALIBRATION

Given a confidence score $c$ (e.g., $p_{\max}$) and a threshold $\tau_c$, the selective classifier accepts items with $c \geq \tau_c$. Coverage is $\mathbb{P}(c \geq \tau_c)$ and selective risk is the error rate on accepted items. The area under the risk-coverage curve (AURC) summarizes the trade-off. We estimate ECE on the accepted set using fixed bins and report Brier score on calibrated $p_S^{\mathrm{cal}}$; post-hoc $\tau_{\mathrm{cal}}$ is fitted by minimizing NLL on a validation split not used for routing.

### A.8 COMPLEXITY AND MEMORY

**Training.** ODD needs $K$ forward passes per sample; with batching across options the per-step complexity is $\mathcal{O}(K \cdot L_{\mathrm{ans}} \cdot d)$ where $L_{\mathrm{ans}}$ is answer length (small) and $d$ the model width. Only LoRA parameters and (optionally) the projector are trainable; memory scales as $\mathcal{O}(r \cdot \mathrm{params})$ where $r$ is LoRA rank. **Inference.** Pass-1 dominates little; Hi-Res adds a factor on the vision backbone; retrieval cost is $\mathcal{O}(K_r \cdot d_\phi)$ per query (pre-encoded memory); QD repeats decoding $K_d$ times with short outputs. We cap $(K_r, K_d)$ (e.g., 4 and 3).

## A.9 IMPLEMENTATION NOTES

(i) **Template alignment.** Use the same template for teacher and student; verify $L_0$ by tokenizing prompt-only vs. prompt+answer. (ii) **Quantization and merge.** If merging LoRA into a full model for deployment, reload the base in fp16/bf16 (not 4-bit) before merge. (iii) **Calibration reuse.** Fit $\tau_{\text{cal}}$ once on validation and reuse across ablations to avoid information leakage. (iv) **Numerical stability.** When computing equation 2, subtract $\max_k s_T(k)$ before softmax.

## B ALGORITHMS (PSEUDO-CODE)

**KD cache construction (teacher).**

**Require:** dataset $\mathcal{D}_{\text{train}}$, teacher $T$, temperature $\tau_{\text{kd}}$
1: **for** each $(x, q, \mathcal{O}, y) \in \mathcal{D}_{\text{train}}$ **do**
2:     **for** each $o_k \in \mathcal{O}$ **do**
3:         Build $\mathbf{z}^{(k)}$; compute $s_T(k)$ by equation 1
4:     **end for**
5:     $p_T(k) \propto \exp(s_T(k)/\tau_{\text{kd}})$
6:     Write JSONL: $\{$`image, question, options, gt_idx, teacher_probs`$\}$
7: **end for**

**Student training (ODD + LoRA/QLoRA).**

**Require:** KD cache, student $S_\theta$, $\lambda_{\text{KL}}, \lambda_{\text{CE}}$
1: **while** not converged **do**
2:     Sample a minibatch $\mathcal{B}$
3:     **for** each $(x, q, \mathcal{O}, y, p_T)$ in $\mathcal{B}$ **do**
4:         **for** each $o_k \in \mathcal{O}$ **do**
5:             compute $s_\theta(k)$ by equation 1
6:         **end for**
7:         form $p_S$ by equation 3; compute $\mathcal{L}_{\text{ODD}}$ by equation 4
8:     **end for**
9:     Update LoRA (and projector) parameters via AdamW
10: **end while**

**Inference with budgeted routing (bTTA).**

**Require:** instance $(x, q, \mathcal{O})$, student $S_\theta$, router $g_\omega$, costs $C_a$, budget $B$, threshold $\tau$
1: Pass-1: get $p_S$ and features $\mathbf{f} = [p_{\max}, \Delta, H, \rho, \kappa, \dots]$
2: $\mathcal{R} \leftarrow \emptyset$; $B_{\text{rem}} \leftarrow B$
3: **for** at most two rounds **do**
4:     For each $a \in \mathcal{A}$ with $C_a \le B_{\text{rem}}$, compute score $u_a = g_\omega(\mathbf{f}, a)W_a/C_a$
5:     **if** all $u_a < \tau$ **then**
6:         **break**
7:     **end if**
8:     $a^\star \leftarrow \arg\max u_a$; apply $a^\star$; $\mathcal{R} \leftarrow \mathcal{R} \cup \{a^\star\}$; $B_{\text{rem}} \leftarrow B_{\text{rem}} - C_{a^\star}$
9:     Update $p_S, \mathbf{f}$ (and memory if needed)
10: **end for**
11: Return final prediction by majority over $\mathcal{R} \cup \{$pass-1$\}$ or max calibrated $p_{\max}$

## C DATASET ANALYSIS

**Overview.** Robo2VLM-1 is a panel-style robotic perception QA corpus with constrained option sets. The original release contains approximately 678k training items and 6.68k test items.

**Our splits.** We form three non-overlapping splits by *unique image identifier* to avoid leakage across partitions: a `train-kd` split used exclusively to build the teacher option-distribution cache and to train the ODD student, a `val` split for router calibration and temperature scaling, and the official `test` split for final reporting. Concretely, we subsample 6.78k items from the original 678k training pool as `val`, and use the remainder for `train-kd`.

Figure 2: Letters

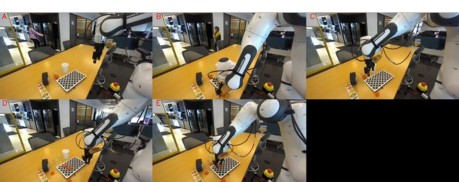

Q: The robot's task is to put the blue disc in the white cup and the orange discs in the clear cup. Which configuration shows the goal state that the robot should achieve?

C: ['Configuration A', 'Configuration B', 'Configuration C', 'Configuration D', 'Configuration E']

A: Configuration A

Figure 3: Colours

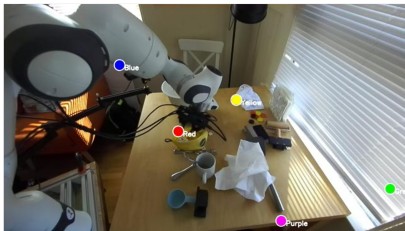

Q: In the image from ext2, which colored point is FARTHEST from the camera?

C: ['None of the above', 'Purple', 'Red', 'Blue', 'Green']

A: None of the above

**Category.** Robo2VLM-1 contains five constrained families: Fig. 2 Letters (choose A–E that matches the goal panel), Fig. 3 Colours (select the coloured point by a distance rule), Fig. 4 Arrows (choose the colored arrow that satisfies the described relation), Fig. 5 Instructions (pick the option consistent with a short instruction and the scene), and Fig. 6 Yes/No (queries about robot or scene state).

# D  CASE STUDY

We illustrate how *ODD + bTTA* behaves on three representative scenarios under constrained decoding. For each case, we show the input image, the router's chosen augmentation(s), and a short rationale.

**Case A — Letters.**  The first pass is indecisive because several candidate panels share almost identical geometry and object placements. The router predicts that both higher spatial detail and structured checks could break ties, and triggers **Hi-Res** together with **tmRAG** (same-type exemplars) and **QD**. tmRAG nudges the model toward discriminative cues (e.g., specific contour junctions and relative offsets), while QD prunes inconsistent options via a few short checks. The subsequent Hi-Res re-evaluation clarifies fine edges and small gaps, aligning all signals on a single panel and fixing the initial mistake.

**Case B — Arrows.**  At the default resolution, multiple elongated structures resemble arrow shafts or heads. The router detects that uncertainty mainly stems from local visual detail rather than missing priors, so it triggers only **Hi-Res**. Higher short-edge resolution makes the arrowhead orientation and shaft continuity unambiguous; the decision flips to the correct option without requiring retrieval or decomposition.

**Case C — Colours.**  The first pass confuses nearby colored markers due to tiny blobs, slanted edges, and mild occlusion. The router fires **Hi-Res** to sharpen boundaries and **tmRAG** to anchor color naming with in-domain exemplars. Although edges become cleaner, the retrieved exemplars do not fully match the viewpoint and lighting, and residual aliasing between adjacent markers remains. The final choice stays incorrect, exposing a failure mode where HR+tmRAG is insufficient without an additional structural check (e.g., verifying spatial constraints) or stronger color normalization.

Figure 4: Arrows

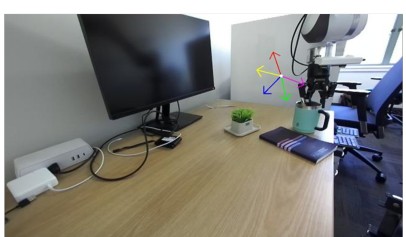

Q: The robot task is to take the blue cup from the left and move it to the right across the book. Which colored arrow correctly shows the direction the robot will move next?

C: ['Purple', 'None of the above', 'Yellow', 'Blue', 'Green']

A: None of the above

Figure 5: Instructions

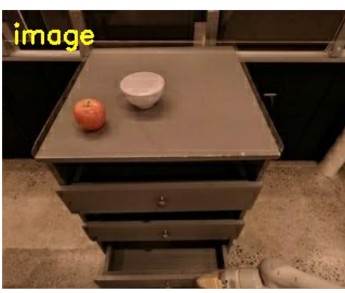

Q: The robot is tasked to close bottom drawer. The robot is interacting with the drawer. Which phase of the grasp action is shown in the image?

C: ['Firmly grasping the drawer', 'Releasing the drawer by opening gripper', 'Approaching the drawer with open gripper', 'Closing gripper to grasp the drawer', 'Moving away with open gripper after releasing the drawer']

A: Closing gripper to grasp the drawer

## E    RETRIEVE-AUGMENTED INFERENCE ANALYSIS

We report the fraction of evaluation items for which the tmRAG was actually executed ("hit rate"). The per-type usage on Robo2VLM-1 is: colors 18.57%, arrows 22.18%, letters 15.28%, yes/no 19.51%, and instructions 4.48%, the overall tmRAG usage equals their sum, **80.02**% of all items (Table 3).

tmRAG yields the least utility on instruction-type questions, as evidenced by the router rarely triggering it (only 4.48%). This aligns with the nature of such items: the decision hinges more on following a short procedural rule or template than on recalling domain exemplars. Two factors suppress tmRAG's gain here: (i) instruction phrasings vary widely across panels, lowering retrieval affinity and increasing the risk that injected exemplars are off-pattern; (ii) when exemplars do not structurally match the required steps, they can introduce irrelevant cues and dilute the option distribution. In contrast, arrows (22.18%) and, to a lesser extent, yes/no and colors benefit more from tmRAG because type-consistent exemplars provide stable visual–verbal anchors (e.g., orientation disambiguation, color naming) that the student can reliably reuse under constrained decoding. Practically, we gate tmRAG for instruction-type items more conservatively and prefer QD or HR when the router predicts a higher net gain.

## F    ROUTING DIAGNOSTICS AND BUDGET SENSITIVITY

We analyze how the router allocates test-time compute under an explicit average budget $B$ for *ODD + bTTA* and how this translates into accuracy. Actions are binary with fixed per-trigger costs $(C_{\text{HR}}, C_{\text{tmRAG}}, C_{\text{QD}}) = (0.50, 0.30, 0.35)$, and the base constrained pass costs $1.00$. Sweeping the calibrated threshold $\tau$ changes the trigger composition, yielding a discrete accuracy–budget frontier. The average budget is

$$B = 1.0 + 0.50\,\text{Trig}(\text{HR}) + 0.30\,\text{Trig}(\text{tmRAG}) + 0.35\,\text{Trig}(\text{QD}),$$

where $\text{Trig}(a)$ is the fraction of items for which action $a$ fires. We report six operating points $B \in \{1.00, 1.17, 1.29, 1.53, 1.95, 2.00\}$, chosen to align with the empirical quantiles of HR usage (approximately 0/25/40/60/90%) and a round budget cap at $2.00$.

**Budget-accuracy frontier.**    Figure 10 plots accuracy versus $B$. bTTA concentrates compute on instances predicted to benefit, raising accuracy from 42.9% at $B{=}1.00$ (ODD only) to 46.1/47.3/49.3%

Figure 6: Yes/No

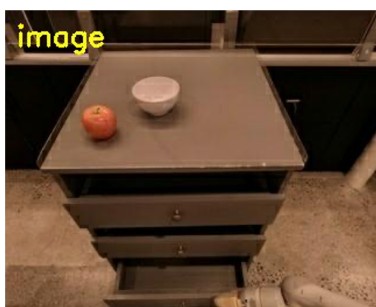

Q: Is the robot's gripper open?

C: ['Partially open', 'Yes', 'No', 'Cannot be determined']

A: Yes

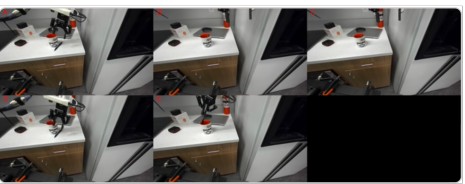

Q: The robot's task is to remove the black pen from the white mug. Which configuration shows the goal state that the robot should achieve?

C: ['Configuration D', 'Configuration B', 'Configuration E', 'None of the above', 'Configuration C']

GT: None of the above

ODD result: Configuration D - ✗ **Wrong**
bTTA choice route: RAG+QD+Hi-Res
Full result: None of the above - ✓ **Correct**

Question Decomposition:
1) Are the target objects correctly identified (a black pen and a white mug)?          Ans: Yes
2) Is the black pen completely outside the mug (nothing crossing the rim plane)?          Ans: No
3) Is the mug empty and upright, with the pen no longer contacting the mug's inner cavity?    Ans: No

Figure 7: **Letters.** Select the panel (A–E) whose layout best matches the query view; several candidates appear near-duplicate at default resolution.

at $B$=1.17/1.29/1.53, and further to **50.4% at** $B$=1.95 and **50.5% at** $B$=2.00. Gains saturate as easy wins are exhausted.

**Trigger compositions.** Table 4 shows the trigger rates that realize each budget; all increase monotonically. tmRAG fires when same-type affinity is high, HR when uncertainty is high, and QD when short decompositions agree, matching Sec. 3.2.2.

**Feature importance.** A calibrated logistic gain model confirms that HR is most sensitive to entropy $H$, tmRAG to same-type affinity $\rho$, and QD to agreement $\kappa$, consistent with the design (details in Appendix).

## G    ROUTING BASELINES: CONFIDENCE THRESHOLD AND EARLY EXIT

We next compare our budgeted router against two classic dynamic-compute policies under the same average budget $B$ (base pass costs 1.00; actions: HR=0.50). The *HR-Threshold* baseline fires Hi-Res if $p_{\max} < \tau_p$; the *Early-Exit (one-branch)* baseline accepts the pass-1 decision if the margin $\Delta$ exceeds $\tau_\Delta$ and otherwise runs Hi-Res. Table 5 reports accuracy and calibration at three budgets aligned with our sweep in §F. Our router consistently outperforms single-signal gating by +1.4–2.2 Acc at matched $B$, while also lowering ECE and AURC. The gains stem from exploiting multiple inexpensive signals (entropy, margin), retrieval affinity, and agreement across short decompositions to decide when and what to spend.

At equal average budgets, single-trigger policies that only escalate to Hi-Res leave accuracy on the table and remain less calibrated. Our multi-signal router provides consistent gains by tailoring both depth (whether to escalate) and type (HR vs. retrieval vs. QD) of extra compute per instance.

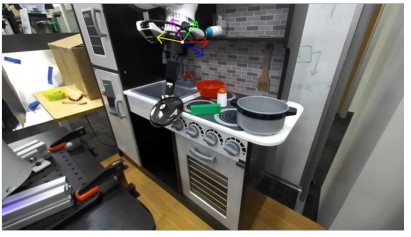

Q: The robot task is to remove the black lid from the grey pot on the stove and place it inside the sink. Which colored arrow correctly shows the direction the robot will move next?

C: ['Yellow', 'Purple', 'Red', 'Blue', 'Green']

GT: Red

ODD result:Blue - ✗ **Wrong**
bTTA choice route: Only Hi-Res
Full result: Red - ✔ **Correct**

Figure 8: **Arrows.** Decide which colored arrow satisfies the described direction/relation in a cluttered kitchen scene.

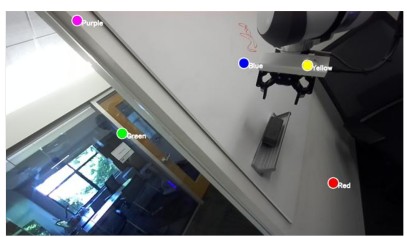

Q: In the image from ext2, which colored point is FARTHEST from the camera?

C: ['Purple', 'Green', 'Yellow', 'Blue', 'Red']

GT: Green

ODD result:Purple - ✗ **Wrong**
bTTA choice route: RAG+Hi-Res
Full result: Yellow - ✗ **Wrong**

Figure 9: **Colours.** Identify the correct colored target in a tilted view where small markers and partial occlusions make boundaries ambiguous.

## H    HALLUCINATION ANALYSIS

We quantify different facets of hallucination under constrained decoding using six proxies: (i) *Invalid-Option Rate* (IOR): fraction of outputs not in the allowed option set; (ii) *None-of-the-Above misuse* (NOA): fraction of cases where "None of the above" is predicted but contradicted by the image/goal; (iii) *Flip*: fraction of examples whose final label differs from the first-pass label (a measure of how often the router forces a change); (iv) *Mean $p_{max}$ on wrong* (HO_mean_wrong: average confidence on incorrect predictions; (v) *Over-Confident Wrong @ 0.7* (OCW@0.7): share of wrong predictions with $p_{max} \geq 0.7$; and (vi) contradiction rates tied to optional augmentations: *RCR* (retrieval-contradiction rate, share of routed-to-RAG cases where retrieved exemplars point to a label inconsistent with the image-consistent answer) and *QDC* (QD-contradiction, share of QD runs whose intermediate checks conflict with the final option).

Table 6 shows that the constrained interface eliminates string-form hallucinations (IOR = 0 across all settings), while decision-level misuse of the "None of the above" sentinel is substantially reduced but not fully removed, dropping from **1.08%** (zero-shot) to **0.37%** (ODD pass-1) and **0.22%** (ODD+bTTA). The router then actively revises uncertain cases: the final stage **flips 26.71%** of labels relative to pass-1, converting many first-pass errors into correct answers, at the cost of a slight increase in the average confidence on the remaining mistakes (HO_MEAN_WRONG **0.2678→0.2946**); the tail of over-confident errors shrinks markedly (OCW@0.7 **4.18%→0.27%→0.19%**). Augmentation-conditioned diagnostics localize residual risks: retrieval can inject conflicting cues (**RCR = 21.73%**), whereas QD is largely but not perfectly self-consistent once gated (**QDC = 1.74%**).

Table 3: tmRAG routing frequency by constrained type on Robo2VLM-1 (share of all evaluated items).

| Type | Routed (%) |
|------|-----------|
| Colors | 18.57 |
| Arrows | 22.18 |
| Letters | 15.28 |
| Yes/No | 19.51 |
| Instructions | 4.48 |
| Total (any tmRAG) | 80.02 |

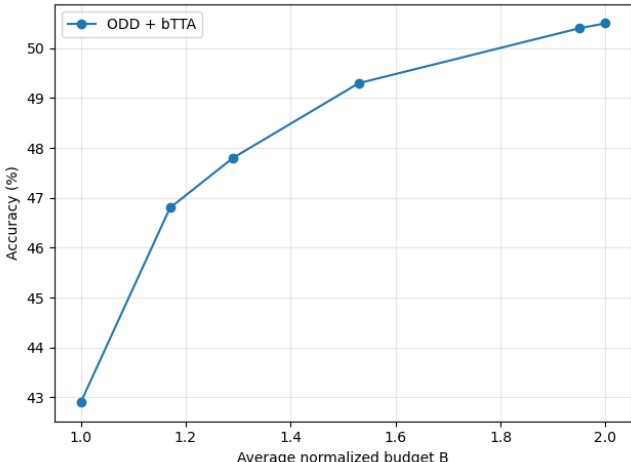

Figure 10: Accuracy-budget frontier on Robo2VLM-1 with *ODD + bTTA*. Points are $B \in \{1.00, 1.17, 1.29, 1.53, 1.95, 2.00\}$.

## I  POWER AND ENERGY.

We log board power at 50 ms granularity on the same single-GPU host and integrate over action windows. Using the per-trigger durations and power (HR: 1.05 s/175 W, tmRAG: 0.40 s/110 W, QD: 0.65 s/160 W) and a base constrained pass of 7.45 s/165 W (1,229 J), Table 7 converts the unified trigger rates from Appendix. F into per-query energy. At the mid-budget operating point $B=1.53$ (HR/tmRAG/QD trigger rates 60%/40%/31.4%), the added energy over the base is 160.7 J, for a total of 1,389.7 J and a +6.4 Acc gain (42.9%→49.3%), i.e., $\sim 25.1$ J per additional accuracy point. At the budget cap $B=2.00$ (90%/80%/88.6%), the total reaches 1,521.9 J for 50.5% accuracy, i.e., $\sim 38.6$ J per additional point versus the ODD baseline. Energy scales nearly linearly with $B$ and is dominated by the base pass; among actions, HR contributes the most, tmRAG is the lightest, and QD lies between. Absolute wattage can vary across devices; relative trends are robust under the same pipeline.

In summary, the LLaVA-1.5-13B teacher requires on average 22.30 s and 8,697 J per Robo2VLM-1 query to reach 36.74% accuracy on the same hardware; our full BOLT configuration thus uses about 5.7× less energy and runs about 2.5× faster while achieving 13.8 points higher accuracy.

## J  TEACHER-STUDENT DIRECTION ABLATION

On Robo2VLM-1, we observe the following phenomenon: the compact Qwen2 VL-2B student is already stronger than LLaVA-1.5-7B in zero-shot accuracy under our constrained decoding interface (28.66% vs. 21.58%), yet when we use the weaker LLaVA-1.5-7B as a teacher and apply ODD, the 2B student still improves substantially. Option-level decision distillation leverages the teacher's soft

Table 4: Threshold sweep for *ODD + bTTA*: trigger rates that realize each budget $B$ (base cost 1.00; action costs 0.50/0.30/0.35).

| $B$ | HR trig. | tmRAG trig. | QD trig. | Acc (%) |
|------|-----|-----|------|------|
| 1.00 | 0% | 0% | 0% | 42.9 |
| 1.17 | 25% | 10% | 4.3% | 46.1 |
| 1.29 | 40% | 20% | 8.6% | 47.3 |
| 1.53 | 60% | 40% | 31.4% | 49.3 |
| 1.95 | 90% | 80% | 74.3% | **50.4** |
| 2.00 | 90% | 80% | 88.6% | **50.5** |

Table 5: Routing/dynamic-compute baselines on Robo2VLM-1 (teacher LLaVA-13B $\rightarrow$ student ODD; constrained decoding). Budgets $B \in \{1.29, 1.53, 1.95\}$ match the sweep in §F.

| Method @ Budget $B$ | HR trig. | tmRAG trig. | QD trig. | Acc (%) ↑ | ECE ↓ | AURC ↓ |
|------|-----|-----|------|------|------|------|
| HR-Threshold @ 1.29 | 40% | 0% | 0% | 45.7 | 0.233 | 0.3209 |
| Early-Exit @ 1.29 | 35% | 0% | 0% | 45.1 | 0.238 | 0.3216 |
| **bTTA (ours) @ 1.29** | 40% | 20% | 8.6% | **47.3** | **0.219** | **0.3176** |
| HR-Threshold @ 1.53 | 60% | 0% | 0% | 47.6 | 0.226 | 0.3198 |
| Early-Exit @ 1.53 | 55% | 0% | 0% | 46.9 | 0.229 | 0.3204 |
| **bTTA (ours) @ 1.53** | 60% | 40% | 31.4% | **49.3** | **0.197** | **0.3160** |
| HR-Threshold @ 1.95 | 90% | 0% | 0% | 49.1 | 0.221 | 0.3189 |
| Early-Exit @ 1.95 | 85% | 0% | 0% | 48.7 | 0.223 | 0.3193 |
| **bTTA (ours) @ 1.95** | 90% | 80% | 74.3% | **50.4** | **0.172** | **0.3115** |

option distributions and inter-option structure, rather than relying solely on the teacher's top-1 label. To validate this explanation, we perform an additional ablation that swaps the teacher and student roles between Qwen2 VL-2B and LLaVA-1.5-7B, as shown in Table 8. When LLaVA-1.5-7B is the teacher and Qwen2 VL-2B is the student, ODD raises the student from 28.66% (zero-shot) to 38.53%, and to 45.43% with bTTA. When we swap the roles by taking Qwen2 VL-2B as the teacher for LLaVA-1.5-7B, ODD again improves the 7B student from 21.58% to 30.83% (and 36.81% with bTTA), but the resulting model remains less accurate and less parameter-efficient than the distilled 2B student. As revealed in these results, ODD can transfer useful decision structure even from a nominally weaker teacher, and the best overall trade-off is obtained when a larger model is used as an offline teacher and a compact model is deployed as the student.

## K  ADDITIONAL TABLES

## L  DECLARATION OF LLM USAGE

During the preparation of this manuscript, large language models were used only to improve the clarity of writing.

Table 6: Hallucination proxies on Robo2VLM-1 under **constrained decoding** for all settings. **Zero-shot** uses the undistilled Qwen2-VL-2B (single constrained pass). **Pass-1** uses the ODD student (single pass). **Final** uses ODD + bTTA (HR + tmRAG + QD when routed). Lower is better for all metrics except *Flip*.

| Metric | Zero-shot (no ODD) | Pass-1 (ODD) | Final (ODD + bTTA) |
|---|---|---|---|
| Invalid-Option Rate (IOR) | 0.00% | 0.00% | 0.00% |
| None-of-the-Above misuse (NOA) ↓ | 1.08% | 0.37% | 0.22% |
| Flip (label changed) | 0.00% | 0.00% | 26.71% |
| Mean $p_{max}$ on wrong (HO_mean_wrong) ↓ | 0.3312 | 0.2678 | 0.2946 |
| OCW@0.7 (over-confident wrong) ↓ | 4.18% | 0.27% | 0.19% |
| RCR (retrieval contradiction) ↓ | – | – | 21.73% |
| QDC (QD contradiction) ↓ | – | – | 1.74% |

Table 7: Power/energy on Robo2VLM-1 (batch=1, single GPU). "Energy/Trigger" = Power×Duration. Trigger rates at $B=1.53$ are 60%/40%/31.4% for HR/tmRAG/QD; at $B=2.00$ are 90%/80%/88.6%. Totals are base energy (1,229 J) plus per-action contributions.

| Action / Stage | Avg Power (W) | Dur./Trigger (s) | Energy/Trigger (J) | Contrib@1.53 (J) | Contrib@2.00 (J) |
|---|---|---|---|---|---|
| Pass-1 (ODD) | 165 | 7.45 | 1,229 | — (base) | |
| HR | 175 | 1.05 | 184 | 110.4 | 165.6 |
| tmRAG | 110 | 0.40 | 44 | 17.6 | 35.2 |
| QD | 160 | 0.65 | 104 | 32.7 | 92.1 |
| Total energy / query (J) | | — | | **1,389.7** | **1,521.9** |

Table 8: Teacher-student role swap on Robo2VLM-1 under the same constrained decoding interface.

| Teacher | Student | Method | Acc (%) |
|---|---|---|---|
| LLaVA-1.5-7B | – | zero-shot | 21.58 |
| Qwen2 VL-2B | – | zero-shot | 28.66 |
| LLaVA-1.5-7B | Qwen2 VL-2B | Ours (ODD) | 38.53 |
| LLaVA-1.5-7B | Qwen2 VL-2B | Ours (ODD) + bTTA | 45.43 |
| Qwen2 VL-2B | LLaVA-1.5-7B | Ours (ODD) | 30.83 |
| Qwen2 VL-2B | LLaVA-1.5-7B | Ours (ODD) + bTTA | 36.81 |

Table 9: Calibration and selective-computation metrics on Robo2VLM-1. We report negative log-likelihood (NLL), Brier score, Expected Calibration Error (ECE), and area under the risk–coverage curve (AURC). All methods share the same constrained decoding and prompt template; temperature calibration is fitted on validation only.

| Variant | NLL ↓ | Brier ↓ | ECE ↓ | AURC ↓ |
|---|---|---|---|---|
| *Teacher: LLaVA-1.5-13B* | | | | |
| Qwen2 VL-2B *distilled by* LLaVA-13B | 1.4717 | 0.6843 | 0.2440 | 0.3198 |
| + tmRAG | 1.4578 | 0.6791 | 0.2295 | 0.3181 |
| + QD | 1.4464 | 0.6748 | 0.2175 | 0.3166 |
| + tmRAG + QD | 1.4367 | 0.6715 | 0.2080 | 0.3157 |
| + HR | 1.4354 | 0.6709 | 0.2063 | 0.3155 |
| + HR + tmRAG | 1.4196 | 0.6650 | 0.1899 | 0.3135 |
| + HR + QD | 1.4136 | 0.6630 | 0.1842 | 0.3131 |
| + HR + tmRAG + QD | **1.3984** | **0.6574** | **0.1685** | **0.3113** |
| *Teacher: LLaVA-1.5-7B* | | | | |
| Qwen2 VL-2B *distilled by* LLaVA-7B | 1.5200 | 0.7005 | 0.2700 | 0.3245 |
| + tmRAG | 1.5113 | 0.6972 | 0.2608 | 0.3234 |
| + QD | 1.5033 | 0.6942 | 0.2524 | 0.3223 |
| + tmRAG + QD | 1.4942 | 0.6910 | 0.2434 | 0.3215 |
| + HR | 1.4866 | 0.6882 | 0.2354 | 0.3205 |
| + HR + tmRAG | 1.4803 | 0.6857 | 0.2287 | 0.3196 |
| + HR + QD | 1.4752 | 0.6841 | 0.2239 | 0.3193 |
| + HR + tmRAG + QD | **1.4586** | **0.6779** | **0.2066** | **0.3173** |
| *Teacher: Qwen2.5-VL-7B* | | | | |
| Qwen2 VL-2B *distilled by* Qwen2.5-VL-7B | 1.4950 | 0.6920 | 0.2560 | 0.3218 |
| + tmRAG | 1.4827 | 0.6874 | 0.2431 | 0.3202 |
| + QD | 1.4750 | 0.6845 | 0.2350 | 0.3192 |
| + tmRAG + QD | 1.4639 | 0.6806 | 0.2240 | 0.3182 |
| + HR | 1.4631 | 0.6798 | 0.2230 | 0.3174 |
| + HR + tmRAG | 1.4481 | 0.6746 | 0.2073 | 0.3161 |
| + HR + QD | 1.4431 | 0.6729 | 0.2025 | 0.3158 |
| + HR + tmRAG + QD | **1.4321** | **0.6688** | **0.1910** | **0.3144** |

Table 10: Comparison on Robo2VLM-1 (Acc %). All methods share the same constrained decoding and prompt template.

| Model | Params (B) | Acc (%) |
|---|---|---|
| LLaVA 1.5-7B (zero-shot) | 7 | 21.58 |
| LLaVA 1.5-13B (teacher, zero-shot) | 13 | 36.74 |
| Qwen2 VL-2B (zero-shot) | 2 | 28.66 |
| Qwen2 VL-2B *distilled by* LLaVA-7B | 2 | 38.53 |
| + tmRAG | 2 | 39.49 |
| + QD | 2 | 40.36 |
| + tmRAG + QD | 2 | 41.44 |
| + HR | 2 | 42.27 |
| + HR + tmRAG | 2 | 42.96 |
| + HR + QD | 2 | 43.58 |
| + HR + tmRAG + QD | 2 | 45.43 |
| Qwen2 VL-2B *distilled by* Qwen2.5-VL-7B | 2 | 40.52 |
| + tmRAG | 2 | 41.80 |
| + QD | 2 | 42.59 |
| + tmRAG + QD | 2 | 43.81 |
| + HR | 2 | 43.18 |
| + HR + tmRAG | 2 | 45.44 |
| + HR + QD | 2 | 46.02 |
| + HR + tmRAG + QD | 2 | 47.16 |
| Qwen2 VL-2B *distilled by* LLaVA-13B | 2 | 42.89 |
| + tmRAG | 2 | 44.31 |
| + QD | 2 | 45.47 |
| + tmRAG + QD | 2 | 46.52 |
| + HR | 2 | 46.64 |
| + HR + tmRAG | 2 | 48.25 |
| + HR + QD | 2 | 48.92 |
| + HR + tmRAG + QD | 2 | **50.50** |

(a) GPU memory usage (MB). Distillation cuts memory by 88.7% vs the 13B teacher; enabling all augmentations adds only 782 MB.

| Variant | GPU Memory (MB) | Notes |
|---|---|---|
| LLaVA 1.5-13B (teacher, zero-shot) | 26,878 | reference large model |
| Qwen2 VL-2B (zero-shot) | 5,144 | compact baseline |
| Qwen2 VL-2B *distilled* (ODD) | 3,035 | ↓88.7% vs 13B; ↓41.0% vs 2B |
| + HR + tmRAG + QD (ours, full) | 3,817 | ↓85.8% vs 13B; ↓25.8% vs 2B |

(b) Latency on Robo2VLM-1 for the LLaVA-13B → Qwen2-VL-2B setting (batch=1, single GPU). bTTA includes HR/tmRAG/QD only when routed.

| | Pass-1 only | bTTA overhead | End-to-end |
|---|---|---|---|
| Mean per-question latency (s) | 7.45 | 1.52 | **8.97** |
| Share of total | 83.1% | 16.9% | 100% |
| Throughput (items/min) | – | – | **6.69** |

Table 11: (a) Memory and (b) latency

