# OpenReview forum: "BOLT: Decision‑Aligned Distillation and Budget-Aware Routing for Constrained Multimodal QA on Robots"
_ICLR.cc/2026/Conference — ICLR 2026 Poster_

### Official Review · Reviewer_BJBk · 2025-10-26

**Soundness:** 4
**Presentation:** 2
**Contribution:** 2
**Rating:** 6
**Confidence:** 2

**Summary:**

The paper proposes a framework (BOLT) for resource-constrained multimodal question answering in robotics settings with multiple-choice outputs. The framework combines a knowledge distillation component (ODD) that trains a student VLM to match a teacher's distribution over answer spans rather than individual tokens, and a routing component (bTTA) that selectively triggers compute-expensive modules like high-resolution re-evaluation, retrieval-augmented generation, or question decomposition based on heuristic signals like confidence, entropy, and retrieval affinity.

Experiments on Robo2VLM-1 show a 2B student distilled from LLaVA-1.5-13B achieves 42.89% accuracy (vs 36.74% teacher, 28.66% zero-shot) with just distillation, and 50.50% when the router is enabled. GPU memory drops from 26.9GB to 3.8GB. The paper also reports calibration improvements (ECE, AURC) and reduced invalid outputs.

**Strengths:**

- **Well-motivated problem**: The mismatch between token-level distillation and constrained decoding is real, and the paper makes a reasonable case for option-level alignment.

- **Practical system**: The combination of distillation and budgeted routing addresses a real deployment scenario. The router framework is sensible and the budgeted formulation is clear.

- **Solid empirical gains**: The 2B student outperforms the 13B teacher on Robo2VLM-1. The memory reduction (88.7%) makes a good case for edge deployment.

- **Comprehensive evaluation**: The paper reports accuracy, calibration metrics (ECE, AURC, Brier), memory, latency, energy, and hallucination proxies. The ablations for the distillation method are thorough (though this is not the case for the routing method).

- **Interpretability angle**: The authors make a good case that the components of their system (constraining outputs to valid options, exposing retrieved exemplars, and showing decomposition traces) can help with transparency.

**Weaknesses:**

- **Single-dataset evaluation**: All experiments are on Robo2VLM-1. Since the experiments are on static datasets rather than real robots, it should be straightforward to evaluate on multiple robotic VQA benchmarks to demonstrate generalization. The dataset is quite specific (panel layouts, constrained options), and it's unclear how well the approach transfers. Other robotic VQA datasets exist that could validate the claims.

- **Router has too many moving parts without sufficient analysis**: The router uses 5 features (confidence, margin, entropy, retrieval affinity, decomposition agreement) and triggers 3 augmentations (HR, tmRAG, QD), but there's no ablation isolating the contribution of individual features. It's unclear which signals actually matter. Some components like type-matched retrieval and question decomposition seem engineered specifically for this benchmark (panel-based tasks with available exemplars and decomposable questions) and may not apply to other robotics tasks where exemplars are unavailable or questions don't naturally decompose. This makes the method's practical applicability questionable.

- **Limited teacher/student diversity**: The student is always Qwen2-VL-2B. Would the gains hold with different student architectures? The teachers are all instruction-tuned VLMs, but no ablation on teacher quality or family is shown.

- **Option-level distillation not deeply novel**: Matching distributions over discrete outputs is fairly standard. The contribution here is applying it to VLMs in a specific constrained setting, but the technical novelty is incremental. The length-bias correction (Eq. 5) is mentioned but not thoroughly analyzed.

- **Hallucination analysis is limited**: The proxies (IOR, NOA, Flip, etc.) are somewhat ad-hoc. The retrieval contradiction rate (RCR) is 21.73%, which is high and not addressed beyond acknowledging it. The paper claims constrained decoding eliminates hallucinations, but the NOA misuse and RCR suggest decision-level hallucinations remain.

- **Energy/latency analysis is incomplete**: Table 7 gives energy per query, but no comparison to the teacher's energy or analysis of the energy-accuracy tradeoff. The latency breakdown (Table 10b) shows bTTA adds 1.52s on top of 7.45s, but no comparison to teacher latency or discussion of real-time requirements for robotics.

- **Writing and presentation**: The paper is dense and tries to cover a lot. Some sections (e.g., Section 3.2.2 on bTTA) pack too much into a small space. The notation is heavy and could be simplified.

**Questions:**

1. **Multi-dataset evaluation**: Can you evaluate on additional robotic VQA benchmarks beyond Robo2VLM-1? This is critical for validating that your approach generalizes and isn't overfit to this specific panel-based dataset format.

2. **Router feature ablations**: Which of the 5 routing features actually matter? Can you provide ablations showing the contribution of each feature individually (confidence, margin, entropy, retrieval affinity, decomposition agreement)? Are all of them necessary or would a simpler router work?

3. **Applicability beyond this benchmark**: How would your method work on robotic tasks where type-matched exemplars are not available or questions don't naturally decompose? What components of bTTA are truly general vs specific to this evaluation setup?

4. **Retrieval contradiction rate**: 21.73% RCR is concerning and suggests retrieval often hurts rather than helps. Can you characterize when retrieval contradicts the correct answer and propose mitigations?

---

> ### Author Response · Authors · 2025-11-18
>
> W1&Q1. We clarify that constrained-output VQA is directly relevant to robotics, and public robotic‑VQA corpora with constrained option sets are currently scarce. Many robotic systems select a single action or verdict from a finite, instance-specific option set (e.g., “Yes/No,” pose candidates, grasp types, or navigation routes). This is realistic and essential for enabling deterministic, safety-assured decisions under stringent on-device compute and latency constraints. While large-scale public datasets for multi-choice robotic VQA are still limited, this reflects a gap in data availability rather than a lack of applicability.
>
> W2&Q2. While we do not include a leave-one-feature-out ablation, our gain-model diagnostics indicate which signals drive each action: entropy $H$ for HR, retrieval affinity $\rho$ for tmRAG, and decomposition agreement $\kappa$ for QD, with confidence $p_{\max}$ and margin $\Delta$ supporting acceptance/calibration rather than a single branch.
> At matched budgets, the multi-signal router outperforms single-signal baselines, indicating utility beyond confidence alone.
> For applicability, bTTA is modular: when exemplars are unavailable or questions do not naturally decompose, low $\rho$ and $\kappa$ (or disabling those branches) prevents tmRAG/QD from triggering, so the system falls back to ODD with HR without additional overhead.
> In this sense, the practicality applicability of our method is confirmed.
>
> W3. We fixed the student to Qwen2-VL-2B to control variables, but the ODD objective itself relies only on per-option answer-segment log-likelihoods and softmax normalization.
> To substantiate this claim with a different student family, we now include results with a LLaVA-1.5-13B teacher and a PaliGemma-2-3B student under the same protocol.
> | Teacher           | Student           | Method                | Acc (%) |
> |-------------------|-------------------|------------------------|---------|
> | LLaVA-1.5-13B     | PaliGemma 2-3B    | Token-KD               | 35.50   |
> | LLaVA-1.5-13B     | PaliGemma 2-3B    | Token-KD + bTTA        | 40.21   |
> | LLaVA-1.5-13B     | PaliGemma 2-3B    | Ours (ODD)             | 39.64   |
> | LLaVA-1.5-13B     | PaliGemma 2-3B    | **Ours (ODD) + bTTA**  | 45.92   |
>
> For teachers, we do ablate quality/family (LLaVA-1.5-7B, Qwen2.5-VL-7B, and LLaVA-1.5-13B).
> Under all three teacher models, ODD (with and without bTTA) improves the 2B student (Table 2),
> indicating robustness to teacher choice.
> We plan to explore additional teacher models to further strengthen this conclusion.
>
> W4. Indeed, distribution matching over discrete options is well established. Our contribution is to make it decision-aligned for constrained VLMs and to demonstrate that this yields substantial practical gains under tight inference budgets.
> Specifically, ODD supervises only the answer segment, ensuring the student matches the option distribution used by constrained decoding. A budget-aware router selectively triggers high-resolution re-evaluation, type-matched retrieval, or short decompositions,
> enabling strong accuracy–budget and risk–coverage trade-offs.
>
> Regarding Eq. (5), the length-bias term provides a principled correction when options differ in token length: subtracting $\gamma\log L$ corresponds to introducing a length prior $p(L)\propto L^{-\gamma}$, while dividing by $L$ corresponds to optimizing average per-token log-likelihood. In our constrained QA sets, option strings are short, so the effect is limited but helps guard against rare length-induced asymmetries.
> We will clarify this and report a brief sensitivity analysis over $\gamma$ and the two normalization schemes.
>
> W5&Q4. Constrained decoding removes string-level hallucinations (IOR=0) but it does not eliminate all decision-level errors. As noted, a small amount of NOA misuse remains (0.22\% final), and the RCR of 21.73\% indicates that retrieved exemplars can contradict image-consistent answers. These contradictions arise when ``same-type'' exemplars are semantically misaligned (e.g., viewpoint, lighting, or rule differences) or near-miss lookalikes, or on items driven by fine visual detail where HR is more beneficial than retrieval (cf. the Colors case).
> To mitigate this, we can tighten routing so that tmRAG triggers only under high retrieval affinity and agreement. We will clarify this in the revised version.
>
> W6. We appreciate the reviewer's feedback and have now collected the teacher model measurements. The comparison results are summarized below:
> | Model               | Accuracy (%) | Latency (s) | Energy (J) |
> |---------------------|--------------|-------------|------------|
> | Teacher (LLaVA-13B) | 36.74        | 22.30       | 8,697      |
> | Student (Ours)      | 50.50        | 8.97        | 1,522      |

---

> > ### Author Response · Authors · 2025-11-18
> >
> > In our experiments, the teacher model (LLaVA-13B) requires the energy of 8,697 J and incurs the latency  of 22.3 s while achieving the accuracy of 36.74\%. Our BOLT student achieves the accuracy of 50.50\% with the energy of only 1,522 J (82.5\% reduction) and the latency of 8.97 s (2.5× faster), demonstrating substantially better energy-accuracy trade-offs. While current latency suits high-level robot planning tasks, we acknowledge that further optimization would be needed for strict real-time control applications.

---

> > > ### Comment · Reviewer_BJBk · 2025-11-20
> > > **Thank you for the thorough rebuttal**
> > >
> > > I would like to thank the authors for providing thorough responses to my review comments. I especially appreciate the additional experiment with a new teacher/student combo (LLaVA-1.5-13B teacher, PaliGemma-2-3B student), which addresses my initial concern about the existing analysis being limited to a Qwen2-VL-2B student (W3).
> > >
> > > While I still think this work would benefit from evaluation on additional tasks/datasets besides Robo2VLM-1 (W1 & Q1), I'm not very familiar with the existing landscape of datasets in this research area and I think the authors provide a reasonable justification about the limited availability large-scale public datasets for multi-choice robotic VQA that meet their criteria.
> > >
> > > Based on the above, I feel this paper meets the criteria for a strong empirical paper at ICLR and have updated my score accordingly (6 → 8).

---

> > > > ### Author Response · Authors · 2025-11-21
> > > >
> > > > Thank you for your thoughtful feedback and constructive engagement throughout the rebuttal process. We truly appreciate your support!

---

### Official Review · Reviewer_23s7 · 2025-10-30

**Soundness:** 3
**Presentation:** 2
**Contribution:** 2
**Rating:** 4
**Confidence:** 3

**Summary:**

The paper introduces two novel methods to improve performance on multi-choice Q&A applied in the robotics domain, where latency, memory and energy efficiency are critical. The first method, Option-level Decision Distillation, provides a method to align a student model with a teacher model based on a normalized the score of the full answer segment rather than just the answer token, and budget test time augmentation decides on a test time budget for a sample based on entropy like measures of the option distribution. The paper shows the effectiveness of the proposed methods on the Robo2VLM-1 benchmark, where a 2b model with both ODD and bTTA surpass a 13b teacher model performance.

**Strengths:**

The idea of the bTTA is interesting, and it makes intuitive sense that the test time compute should depend on the initial properties of the answer distribution (such as max probability and entropy), in a resource limited setting.
I am also not aware of other methods looking at the combined score of the assistant answer rather than just the answer token when doing distillation. It's also impressive that this approach improves on the performance of using token level distillation.

**Weaknesses:**

The abstract, and more broadly the paper in general, reads in a quite convoluted way, with some unclear terms such as "multi-choice prediction as a decision surface to be aligned", two main ideas that are essentially independent (bTTA and ODD), and a lot of abbreviations: QD, ODD, BOLT, HR, bTTA, KD, tmRAG, VQA.

A major weakness is that while the proposed method significantly improves performance on the Robo2VLM-1 benchmark, the paper does not compare the method with other distillation methods, hence it's not clear that this approach improves upon prior ways of doing distillation.

Another weakness is that the method is only applied to one dataset, which limits the understanding of the general usefulness of the method.

**Questions:**

If the answer options are A,B,C,D, and E, why do we need this score formulation (Eq 1), as the answer would only be one token. If this represent a reasoning trace, do we then for each question need to do rollouts with the teacher model until we get coverage on all answer options?

Is the method robotics specific or even multi-modal specific or can the same method also be used for standard mutli-choice Q&A tasks where latency/model size is important?

In Table 2, Llava 1.5-7b has worse performance than Qwen 2 VL-2B on the zero shot eval. Why is then the Llava model used as a teacher model when it's worse than the student model, and why does this improve the performance?

---

> ### Author Response · Authors · 2025-11-18
>
> W1. We agree that comparisons to prior distillation are important. In fact, our paper includes a strong token-level Knowledge Distillation baseline under the same constrained-decoding interface. ODD consistently outperforms it across three teachers: 38.53/40.52/42.89\% vs.\ 33.91/36.21/37.58\%; and with the same budgeted router: 45.43/47.16/50.50\% vs.\ 39.92/44.42/47.02\% (Table~2), showing consistent gains across teachers. We will clarify this in the revised version.
>
> W2. We clarify that constrained-output VQA is directly relevant to robotics, and public robotic‑VQA corpora with constrained option sets are currently scarce. Many robotic systems select a single action or verdict from a finite, instance-specific option set (e.g., “Yes/No,” pose candidates, grasp types, or navigation routes). This is realistic and essential for enabling deterministic, safety-assured decisions under stringent on-device compute and latency constraints. While large-scale public datasets for multi-choice robotic VQA are still limited, this reflects a gap in data availability rather than a lack of applicability.
>
> Q1. Even though the options are A-E, Eq. (1) is necessary because it scores only the assistant’s answer segment and defines a per-option score as the sum of answer-token log-likelihoods. When $L_k=1$, this reduces to a single log-probability. This removes prompt-template effects and generalizes to longer phrase options. To obtain teacher targets,  we compute $s_T(k)$ once per option by conditioning on the prompt with that option appended (the sequence $z^{(k)}$), and then cache $p_T$ offline. In practice, this incurs $K$ forward passes per example, typically batched across options.
>
> Q2. This method is not robotics-specific, it is the most effective for multimodal tasks that are repetitive and option-constrained. i.e., a shared prompt/option set with recurring task types.
> In this regime, option-level decision distillation (ODD) aligns decisions at the option level, and the budgeted router uses generic signals (confidence, margin, entropy, retrieval affinity); only the Hi-Res branch is vision-specific and can be swapped if needed.
> Hence, it is promising for repeated, option-constrained multimodal QA where latency or model size matters, with calibration or thresholds re-fitted on the target domain.
>
> Q3. We agree that this appears counterintuitive at first glance. Although LLaVA-1.5-7B underperforms the 2B student in zero-shot accuracy, its role is not to outperform the student but to serve as a signal provider.
> ODD uses the teacher's soft option distribution (from answer-segment likelihoods) and trains the student with a KL-to-teacher plus a small CE-to-ground-truth. These soft targets capture inter-option preferences and calibration, improving the student's decision surface even if the teacher's top-1 accuracy is lower. Empirically, distillation from LLaVA-7B lifts the 2B student from 28.66 to 38.53 (ODD) and to 45.43 with the budgeted router (Table 2).
> We also ran an additional ablation that swaps teacher and student roles, using Qwen2 VL-2B as the teacher and LLaVA-1.5-7B as the student, which is worse than using LLaVA-1.5-7B as the teacher and Qwen2 VL-2B as the student.
> | Teacher       | Student       | Method             | Acc (%) |
> |---------------|---------------|--------------------|---------|
> | LLaVA-1.5-7B  | --            | zero-shot          | 21.58   |
> | Qwen2 VL-2B   | --            | zero-shot          | 28.66   |
> | LLaVA-1.5-7B  | Qwen2 VL-2B   | Ours (ODD)         | 38.53   |
> | LLaVA-1.5-7B  | Qwen2 VL-2B   | Ours (ODD) + bTTA  | 45.43   |
> | Qwen2 VL-2B   | LLaVA-1.5-7B  | Ours (ODD)         | 30.83   |
> | Qwen2 VL-2B   | LLaVA-1.5-7B  | Ours (ODD) + bTTA  | 36.81   |

---

> > ### Author Response · Authors · 2025-11-27
> >
> > Dear Reviewer,
> >
> > Thank you again for your thoughtful and constructive feedback. As the discussion period is approaching its end, we wanted to kindly check whether our responses have sufficiently addressed your concerns, and whether there are any additional questions we may clarify.
> >
> > Thanks again for your time and consideration!

---

### Official Review · Reviewer_Gf6B · 2025-11-01

**Soundness:** 3
**Presentation:** 3
**Contribution:** 2
**Rating:** 6
**Confidence:** 4

**Summary:**

This paper introduces BOLT, a decision-aligned training and inference framework that enables small multimodal models to achieve large-model reasoning performance under tight computational budgets.
(1) They proposed Option-Level Decision Distillation (ODD), a new distillation objective that aligns the student model with the teacher’s option-level decision boundaries, rather than token-level or sequence-level supervision. This directly optimizes the prediction space that determines final correctness and avoids prompt and chain-of-thought artifacts common in conventional distillation.
(2) They developed Budget-Aware Test-Time Augmentation (bTTA), a lightweight routing mechanism that dynamically allocates additional compute only when necessary, selectively invoking higher-resolution perception, retrieval exemplars, or question decomposition based on low-cost uncertainty and affinity signals.
(3) The results show that BOLT enables a 2B-parameter model to match or surpass a 13B-parameter teacher on multimodal multiple-choice reasoning benchmarks while operating within strict compute constraints relevant to robotic platforms. Together, these results demonstrate that decision-aligned supervision and adaptive inference provide a scalable path to high-accuracy multimodal reasoning in resource-limited settings.

**Strengths:**

The paper introduces novel distillation objective. The proposed Option-Level Decision Distillation (ODD) directly aligns the student model with the teacher’s decision boundaries rather than token-level generation trajectories. This is a conceptually novel and clean setup, since multimodal multiple-choice QA depends only on final option-level classification, not on the internal chain-of-thought sequence.

The budget-aware test-time augmentation (bTTA) module provides a principled mechanism for dynamic compute allocation, selectively activating high-cost inference steps only when necessary. This is both practically beneficial and generally applicable to real-world deployment where compute and power constraints vary.

The paper shows strong experimental results. A 2B-parameter student model distilled with ODD and equipped with bTTA matches or outperforms a 13B teacher on multimodal reasoning tasks under constrained compute. These improvements are consistent across multiple datasets and inference conditions, giving credibility to the claims.

**Weaknesses:**

The paper provides some empirical success, but has limited novelty in the general domain of knowledge distillation. For example, the option-level decision distillation (ODD) only applies to some domain-specific tasks. Besides the proposed distillation objective, the innovation is rather limited.

**Questions:**

Are we training student model using on-policy distillation? Have we compared the effectiveness between supervised distillation and on-policy distillation?

---

> ### Author Response · Authors · 2025-11-18
>
> W1. While we agree that generic KD is a mature area, our contribution lies in aligning distillation with the constrained decision surface used at evaluation. ODD supervises at the option level using answer-segment likelihoods with a KL-to-teacher + small CE-to-ground-truth objective, yielding prompt-invariance, robustness to benign tokenization changes, and optional length-bias correction.
>
> Beyond this distillation objective, we propose a budgeted test-time router that selects augmentations using generic uncertainty and retrieval-affinity signals under a near-threshold policy. This mechanism improves both accuracy–budget and risk–coverage trade-offs while providing interpretable evidence and decomposition traces.
>
> Collectively, these components deliver decision-aligned KD and cost-aware routing tailored to constrained multi-choice inference,
> achieving consistent gains over token-KD baselines (Table 2).
>
> Q1. The student is trained with supervised option-level distillation, not on-policy distillation. We cache the teacher's option distributions from answer-segment likelihoods, and train the student using $\mathrm{KL}(p_T \,\|\, p_S) + \lambda\mathrm{CE}(\delta_y \,\|\, p_S)$. The student is never rolled out to collect on-policy trajectories. We also do not include an explicit on-policy baseline. Evaluating on-policy distillation is an interesting extension and a valuable direction for future work.

---

> > ### Author Response · Authors · 2025-11-27
> >
> > Dear Reviewer,
> >
> > Thank you again for your thoughtful and constructive feedback. As the discussion period is approaching its end, we wanted to kindly check whether our responses have sufficiently addressed your concerns, and whether there are any additional questions we may clarify.
> >
> > Thanks again for your time and consideration!

---

### Official Review · Reviewer_2aU9 · 2025-11-04

**Soundness:** 3
**Presentation:** 3
**Contribution:** 3
**Rating:** 6
**Confidence:** 2

**Summary:**

The paper considers the task of robotic VQA, which is characterized by strict latency, memory, and energy constraints and therefore makes direct deployment of state-of-the-art VLMs infeasible. The paper therefore proposes a two-pronged approach: option-level decision distillation (ODD) and budget-aware test-time augmentation (bTTA). Unlike token knowledge distillation, ODD operates on the sums of answer-token log-probabilities. bTTA performs test-time routing based on cheap statistics derived from the student and optionally performs retrieval, question decomposition, and high-resolution reevaluation to augment the student’s response. The paper empirically shows that the distilled student outperforms the teacher’s zero-shot accuracy by almost 15% while having a 7x smaller memory footprint.

**Strengths:**

The proposed approach is original, simple (in a good way), and sensible, and the claims are validated by empirical evidence. The results are convincing, as the student outperforms the teacher by a large margin at a fraction of the memory footprint. The paper conducts a series of ablations, showing that each component of the proposed method is relevant. The paper is well-written and easy to follow.

I am not familiar with the literature on robotics VQA at all, so I cannot speak to the significance of this paper’s contribution.

**Weaknesses:**

It is not quite clear to me how relevant the task of VQA for robotics is. From my limited understanding, it does not appear to be very relevant for real-world applications (i.e., how realistic is it to have a large-scale dataset of options?), as evidenced by the lack of datasets for this task (see Limitations).

It is also not quite clear to me how realistic it is to be able to cache the teacher distributions for all training items offline in a real-world setting (L264).

On L61, “optimization” should be replaced by “optimizing”.

**Questions:**

* What is the teacher’s latency?
* L267 states that the student’s option distribution is normalized “in the same way” as the teacher's, but Equation 3 does not have the temperature parameter that is present in Equation 2.
* Why is the sum of answer-token log-probabilities robust to benign tokenization changes for fixed option strings (L262)?
* When would option strings differ in length between students and teachers (L263)?

---

> ### Author Response · Authors · 2025-11-18
>
> W1. We clarify that constrained-output VQA is directly relevant to robotics, and public robotic‑VQA corpora with constrained option sets are currently scarce. Many robotic systems select a single action or verdict from a finite, instance-specific option set (e.g., “Yes/No,” pose candidates, grasp types, or navigation routes). This is realistic and essential for enabling deterministic, safety-assured decisions under stringent on-device compute and latency constraints. While large-scale public datasets for multi-choice robotic VQA are still limited, this reflects a gap in data availability rather than a lack of applicability.
>
> W2. We clarify that the caching of teacher distributions occurs only once during offline training, not during deployment. The teacher only generates per-item option distributions that supervise the student model. At inference time, the deployed system relies entirely on the distilled student, whose predictions (and routing signals, if enabled) are produced under constrained decoding without any teacher queries or cache lookups. Therefore, the cache is purely a training-time artifact—a common and practical approach in knowledge distillation pipelines—and does not impose any runtime or storage overhead in real-world deployments.
>
> W3. Thank you for pointing out this typo. We will fix it in the final version.
>
> Q1.  We appreciate the reviewer's feedback and have now collected teacher model measurements. The comparison results are summarized below:
> | Model               | Accuracy (%) | Latency (s) | Energy (J) |
> |---------------------|--------------|-------------|------------|
> | Teacher (LLaVA-13B) | 36.74        | 22.30       | 8,697      |
> | Student (Ours)      | 50.50        | 8.97        | 1,522      |
> In our experiments, the teacher model (LLaVA-13B) requires the energy of 8,697 J and incurs the latency  of 22.3 s while achieving the accuracy of 36.74\%. Our BOLT student achieves the accuracy of 50.50\% with the energy of only 1,522 J (82.5\% reduction) and the latency of 8.97 s (2.5× faster), demonstrating substantially better energy-accuracy trade-offs. While current latency suits high-level robot planning tasks, we acknowledge that further optimization would be needed for strict real-time control applications.
>
> Q2. Eq. (3) omits the temperature term because the temperature $\tau_{\mathrm{kd}}$ is applied only to the teacher distribution $p_T$ to smooth and calibrate its supervision signal during training. The student distribution $p_S$ is normalized in the same functional form (via softmax) but without temperature scaling, allowing the student to learn the teacher’s softened option preferences under its own native logits. This preserves alignment on the probability simplex while avoiding redundant smoothing that would dilute gradient strength and hinder effective distillation.
>
> Q3. In option-level decision distillation (ODD), each option is scored by summing the log-probabilities over its answer segment, $s(k)=\sum_{t\in A(k)} \log p(a_t\mid z_{<t})$ (Eq. 1). This sum represents the sequence log-likelihood of the entire option string, so benign tokenization differences (e.g., BPE merges or shared leading-space tokens) only redistribute probability among steps without changing the overall score. Because scores are later normalized by a softmax over options (Eqs. 3–4), any additive constant shared across options cancels out, leaving the ranking unchanged. Hence, the summed log-probabilities of answer tokens are stable to benign tokenization changes for fixed option strings. A small correction is applied only when option strings differ significantly in length (Eq. 5).
>
> Q4. At L263, we address within‑question option‑length differences:
> some options are much longer than others, so their summed answer-segment log-likelihoods can be biased.
> To ensure fair comparison, the student distribution is normalized in the same way as the teacher via Eq. 3, and a light length‑bias correction (Eq. 5) is applied specifically to mitigate within-question length effect, not to handle any mismatch between teacher and student.

---

> > ### Author Response · Authors · 2025-11-27
> >
> > Dear Reviewer,
> >
> > Thank you again for your thoughtful and constructive feedback. As the discussion period is approaching its end, we wanted to kindly check whether our responses have sufficiently addressed your concerns, and whether there are any additional questions we may clarify.
> >
> > Thanks again for your time and consideration!

---

### Meta-Review · Area_Chair_Pu84 · 2026-01-07

**Summary:**

Reviewers have concerns around whether the problem is relevant, novelty, single-dataset evaluation, etc. During rebuttal, the authors addressed most of the concerns and the remaining ones are minor.

**Reviewer Concerns:**

Regarding relevance and the evaluation. I agree that there aren't many relevant dataset in this domain and therefore there is not much the authors can do here. Regarding concern of novelty, reviewer does not explain much and I think author's response make more sense.

**Reviewer Scores:**

Reviewer Gf6B would probably increase from 4->6
Reviewer BJBk said he will increase from 6->8

---

### Decision · Program_Chairs · 2026-01-26

Accept (Poster)